# DIGRAF: Diffeomorphic Graph-Adaptive Activation Function

**Krishna Sri Ipsit Mantri**[*]
Purdue University
mantrik@purdue.edu

**Xinzhi (Aurora) Wang**[*]
Purdue University
wang6171@purdue.edu

**Carola-Bibiane Schönlieb**
University of Cambridge
cbs31@cam.ac.uk

**Bruno Ribeiro**
Purdue University
ribeirob@purdue.edu

**Beatrice Bevilacqua**[†]
Purdue University
bbevilac@purdue.edu

**Moshe Eliasof**[†]
University of Cambridge
me532@cam.ac.uk

## Abstract

In this paper, we propose a novel activation function tailored specifically for graph data in Graph Neural Networks (GNNs). Motivated by the need for graph-adaptive and flexible activation functions, we introduce DIGRAF, leveraging Continuous Piecewise-Affine Based (CPAB) transformations, which we augment with an additional GNN to learn a graph-adaptive diffeomorphic activation function in an end-to-end manner. In addition to its graph-adaptivity and flexibility, DIGRAF also possesses properties that are widely recognized as desirable for activation functions, such as differentiability, boundness within the domain, and computational efficiency. We conduct an extensive set of experiments across diverse datasets and tasks, demonstrating a consistent and superior performance of DIGRAF compared to traditional and graph-specific activation functions, highlighting its effectiveness as an activation function for GNNs. Our code is available at https://github.com/ipsitmantri/DiGRAF.

## 1 Introduction

Graph Neural Networks (GNNs) have found application across diverse domains, including social networks, recommendation systems, bioinformatics, and chemical analysis [82, 91, 68]. Recent advancements in GNN research have predominantly focused on exploring the design space of key architectural elements, ranging from expressive GNN layers [57, 23, 88, 89, 65], to pooling layers [87, 46, 5, 80], and positional and structural encodings [18, 67, 20]. Despite the exploration of these architectural choices, a common trend persists where most GNNs default to employing standard activation functions, such as ReLU [26], among a few others.

Activation functions play a crucial role in neural networks, as they are necessary for modeling non-linear input-output mappings. Importantly, different activation functions exhibit distinct behaviors, and the choice of the activation function can significantly influence the performance of the neural network [60]. It is well-known [61, 72] that, from a theoretical point of view, non-convex and highly oscillatory activation functions offer better approximation power. However, due to their strong non-convexity, they amplify optimization challenges [39].

Therefore, as a middle-ground between practice and theory, it has been suggested that a successful activation function should possess the following properties: (1) be differentiable everywhere [16, 54], (2) have non-vanishing gradients [16]; (3) be bounded to improve the training stability [48, 16]; (4) be zero-centered to accelerate convergence [16]; and (5) be efficient and not increase the complexity of the neural network [45]. In the context of graph data, the activation function should arguably also be what we define as *graph-adaptive*, that is, tailored

---

[*]Equal contribution.
[†]Equal supervision.

38th Conference on Neural Information Processing Systems (NeurIPS 2024).

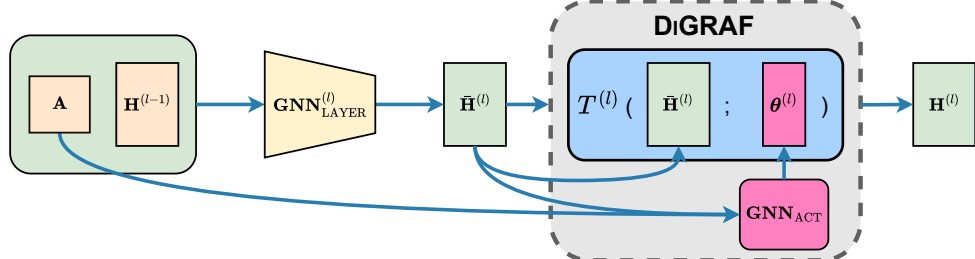

Figure 1: Illustration of DIGRAF. Node features $\mathbf{H}^{(l-1)}$ and adjacency matrix $\mathbf{A}$ are fed to a $\mathrm{GNN}^{(l)}_{\mathrm{LAYER}}$ to obtain updated intermediate node features $\bar{\mathbf{H}}^{(l)}$, which are passed to our activation function layer, DIGRAF. First, an additional GNN network $\mathrm{GNN}_{\mathrm{ACT}}$ takes $\bar{\mathbf{H}}^{(l)}$ and $\mathbf{A}$ as input to determine the activation function parameters $\boldsymbol{\theta}^{(l)}$. These are used to parameterize the transformation $T^{(l)}$, which operates on $\bar{\mathbf{H}}^{(l)}$ to produce the activated node features $\mathbf{H}^{(l)}$.

to the input graph and capable of capturing the unique properties of graph-structured data, such as degree differences or size changes. This adaptivity ensures that the activation function can effectively leverage the structural information present in the graph data, potentially leading to improved performance in graph tasks.

Recent work in graph learning has investigated the impact of activation functions specifically designed for graphs, such as Iancu et al. [38] that proposes graph-adaptive max and median activation filters, and Zhang et al. [92] that introduces GReLU, which learns piecewise linear activation functions with a graph-adaptive mechanism. Despite the potential demonstrated by these approaches, the proposed activation functions still have predefined fixed structures (max and median functions in Iancu et al. [38] and piecewise linear in Zhang et al. [92]), restricting the flexibility of the activation functions that can be learned. Additionally, in the case of GReLU, the learned activation functions inherit the drawback of points of non-differentiability, which are undesirable according to the properties mentioned above. As a consequence, to the best of our knowledge, none of the existing activation functions prove to be consistently beneficial across different graph datasets and tasks. Therefore, *our objective is to design a flexible activation function tailored for graph data, offering consistent performance gains*. This activation function should possess many, if not all, of the properties recognized as beneficial for activation functions, with an emphasis on blueprint flexibility, as well as task and input adaptivity.

**Our Approach: DIGRAF.** In this paper, we leverage the success of learning diffeomorphisms, particularly through Continuous Piecewise-Affine Based transformations (CPAB) [24, 25], to devise an activation function tailored for graph-structured data. Diffeomorphisms, characterized as bijective, differentiable, and invertible mappings with a differentiable inverse, inherently possess many desirable properties of activation functions, like differentiability, boundedness within the input-output domain, and stability to input perturbations. To augment our activation function with graph-adaptivity, we employ an additional GNN to derive the parameters of the learned diffeomorphism. This integration yields our node permutation equivariant activation function, dubbed DIGRAF – **DI**ffeomorphism-based **GR**aph **A**ctivation **F**unction, illustrated in Figure 1, that dynamically adapts to different graphs, providing a flexible framework capable of learning activation functions for specific tasks and datasets in an end-to-end manner. This comprehensive set of characteristics positions DIGRAF as a promising approach for designing activation functions for GNNs.

To evaluate the efficacy of DIGRAF, we conduct an extensive set of experiments on a diverse set of datasets across various tasks, including node classification, graph classification, and regression. Our evaluation compares the performance of DIGRAF with three types of baselines: traditional activation functions, activation functions with trainable parameters, and graph activation functions. Our experimental results demonstrate that DIGRAF repeatedly exhibits better downstream performance than other approaches, reflecting the theoretical understanding and rationale underlying its design and the properties it possesses. Importantly, while existing activation functions offer different behavior in different datasets, DIGRAF maintains consistent performance across diverse experimental evaluations, further highlighting its effectiveness.

**Main contributions.** The contributions of this work are summarized as follows: (1) We introduce a learnable graph-adaptive activation function based on flexible and efficient diffeomorphisms – DIGRAF, which we show to have properties advocated in literature; (2) an analysis of such properties, reasoning about the design choices of our method; and, (3) a comprehensive experimental evaluation of DIGRAF and other activation functions.

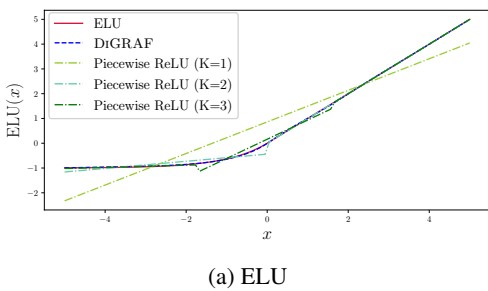
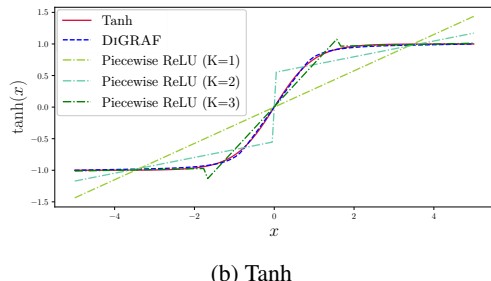

|(a) ELU|(b) Tanh|

Figure 2: Approximation of traditional activation functions using CPAB and Piecewise ReLU with varying segment counts $K \in \{1, 2, 3\}$ on a closed interval $\Omega = [-5, 5]$, demonstrating the advantage of utilizing CPAB and its flexibility to model various activation functions.

## 2 Related Work

**Diffeomorphisms in Neural Networks.** A bijection mapping function $f : \mathcal{M} \to \mathcal{N}$, given two differentiable manifolds $\mathcal{M}$ and $\mathcal{N}$, is termed a *diffeomorphism* if its inverse $f^{-1} : \mathcal{N} \to \mathcal{M}$ is also differentiable. The challenge in learning diffeomorphisms arises from their computational complexity: early research is often based on complicated infinite dimensional spaces [76], and later advancements have turned to Markov Chain Monte Carlo methods, which still suffer from large computational complexity [1, 2, 90]. To address these drawbacks, Freifeld et al. [24, 25] introduced the Continuous Piecewise-Affine Based transformation (CPAB) approach, offering a more pragmatic solution to learning diffeomorphisms by starting from a finite-dimensional space, and allowing for exact diffeomorphism computations in the case of 1D diffeomorphisms – an essential trait in our case, given that activation functions are 1D functions. CPAB has linear complexity and is parallelizable, which can lead to sub-linear complexity in practice [25]. Originally designed for alignment and regression tasks by learning diffeomorphisms, in recent years, CPAB was found to be effective in addressing numerous applications using neural networks, posing it as a suitable framework for learning transformation. For instance, Detlefsen et al. [15] learns CPAB transformations to improve the flexibility of spatial transformer layers, Martinez et al. [52] combines CPAB with neural networks for temporal alignment, Weber and Freifeld [81] introduces a novel loss function that eliminates the need for CPAB deformation regularization in time-series analysis, and Wang et al. [79] utilizes CPAB to model complex spatial transformation for image animation and motion modeling.

**General-Purpose Activation Functions.** In the last decades, the design of activation functions has seen extensive exploration, resulting in the introduction of numerous high-performing approaches, as summarized in Dubey et al. [16], Kunc and Kléma [45]. The focus has gradually shifted from traditional, static activation functions such as ReLU [26], Sigmoid [45], Tanh [35], and ELU [11], to learnable functions. In the landscape of learnable activation functions, the Maxout [29] unit selects the maximum output from learnable linear functions, and PReLU [33] extends ReLU by learning a negative slope. Additionally, the Swish function [66] augments the SiLU function [19], a Sigmoid-weighted linear unit, with a learnable parameter controlling the amount of non-linearity. The recently proposed AdAct [51] learns a weighted combination of several activation functions, and DiTAC [9] learns a diffeomorphic activation function for CNNs. However, these activation functions are not input-adaptive, a desirable property in GNNs.

**Graph Activation Functions.** Typically, GNNs are coupled with conventional activation functions [43, 78, 84], which were not originally tailored for graph data, graph tasks, or GNN models. This implies that these activation functions do not inherently adapt to the structure of the input graph, which was found to be an important property in other GNN components, such as graph normalization [21]. Recent works have suggested various approaches to bridge this gap. Early works such as Scardapane et al. [70] propose learning activation functions based on graph kernels, and Iancu et al. [38] introduces Max and Median filters, which operate on local neighborhoods in the graph, thereby offering adaptivity to the input graphs. A notable advancement in graph-adaptive activation functions is GReLU [92], a parametric piecewise affine activation function achieving graph adaptivity by learning parameters through a hyperfunction that takes into account the node features and the connectivity of the graph. While these approaches demonstrate the potential to enhance GNN performance compared to standard activation functions, they are constrained by their blueprint, often relying on piecewise ReLU composition, which can be performance-limiting [41]. Moreover, a fixed blueprint limits flexibility, i.e., the ability to express a variety of functions. As we show in Figure 2, attempts to approximate traditional activation functions such as ELU and Tanh using piecewise ReLU composition with different segment counts ($K = 1$, 2, and 3), reveal limited approximation power. On the contrary, our DiGRAF, which leverages CPAB, yields significantly better approximations. Furthermore, we demonstrate the approximation power of activations learned with the CPAB framework in our DiGRAF in Appendix E.1.

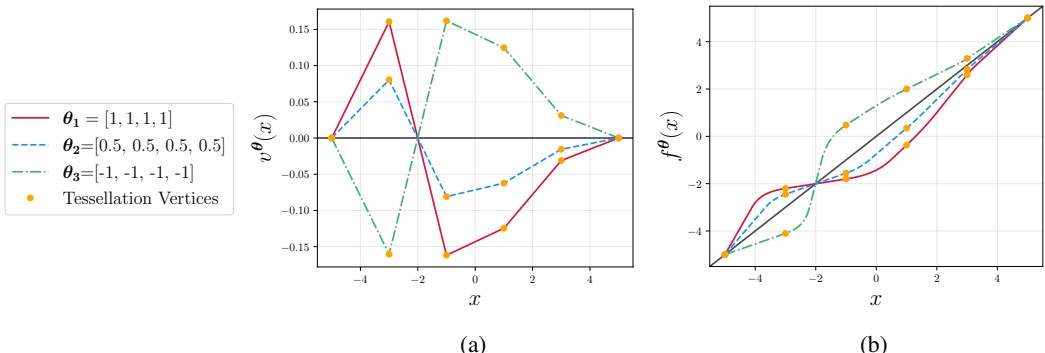

Figure 3: An example of CPA velocity fields $v^{\boldsymbol{\theta}}$ defined on the interval $\Omega = [-5, 5]$ with a tessellation $\mathcal{P}$ consisting of five subintervals. The three different parameters, $\boldsymbol{\theta}_1$, $\boldsymbol{\theta}_2$, and $\boldsymbol{\theta}_3$ define three distinct CPA velocity fields (Figure 3a) resulting in separate CPAB diffeomorphisms $f^{\boldsymbol{\theta}}(x)$ (Figure 3b).

## 3 Mathematical Background and Notations

In this paper, we utilize the definitions from CPAB — a framework for efficiently learning flexible diffeomorphisms [24, 25], alongside basic graph learning notations, to develop activation functions for GNNs. Consequently, this section outlines the essential details needed to understand the foundations of our DIGRAF.

### 3.1 CPAB Diffeomorphisms

Let $\Omega = [a, b] \subset \mathbb{R}$ be a closed interval, where $a < b$. We discretize $\Omega$ using a tessellation $\mathcal{P}$ with $\mathcal{N}_{\mathcal{P}}$ intervals, which, in practice, is oftentimes an equispaced 1D meshgrid with $\mathcal{N}_{\mathcal{P}}$ segments [25] (see Appendix C for a formal definition of tessellation). Our goal in this paper is to learn a diffeomorphism $f : \Omega \to \Omega$ that we will use as an activation function. Formally, a diffeomorphism is defined as follows:

**Definition 3.1** (Diffeomorphism on a closed interval $\Omega$). A diffeomorphism on a closed interval $\Omega \subset \mathbb{R}$ is any function $f : \Omega \to \Omega$ that is (1) bijective, (2) differentiable, and (3) has a differentiable inverse $f^{-1}$.

To instantiate a CPAB diffeomorphism $f$, we define a continuous piecewise-affine (CPA) velocity field $v^{\boldsymbol{\theta}}$ parameterized by $\boldsymbol{\theta} \in \mathbb{R}^{\mathcal{N}_{\mathcal{P}}-1}$. We display examples of velocity fields $v^{\boldsymbol{\theta}}$ for various instances of $\boldsymbol{\theta}$ in Figure 3a to demonstrate the distinct influence of $\boldsymbol{\theta}$ on $v^{\boldsymbol{\theta}}$. Formally, a velocity field $v^{\boldsymbol{\theta}}$ is defined as follows:

**Definition 3.2** (CPA velocity field $v^{\boldsymbol{\theta}}$ on $\Omega$). Given a tessellation $\mathcal{P}$ with $\mathcal{N}_{\mathcal{P}}$ intervals on a closed domain $\Omega$, any velocity field $v^{\boldsymbol{\theta}} : \Omega \to \mathbb{R}$ is termed continuous and piecewise-affine if (1) $v^{\boldsymbol{\theta}}$ is continuous, and (2) $v^{\boldsymbol{\theta}}$ is an affine transformation on each interval of $\mathcal{P}$.

The CPA velocity field $v^{\boldsymbol{\theta}}$ defines a differentiable trajectory $\phi^{\boldsymbol{\theta}}(x, t) : \Omega \times \mathbb{R} \to \Omega$ for each $x \in \Omega$. The trajectories are computed by integrating the velocity field $v^{\boldsymbol{\theta}}$ to time $t$, and are used to construct the CPAB diffeomorphism. We visualize the resulting diffeomorphism in Figure 3b with matching colors denoting corresponding pairs of $v^{\boldsymbol{\theta}}$ and $f^{\boldsymbol{\theta}}(x)$. Mathematically,

**Definition 3.3** (CPAB Diffeomorphism). Given a CPA velocity field $v^{\boldsymbol{\theta}}$, the CPAB diffeomorphism $f$ at point $x$, is defined as:

$$f^{\boldsymbol{\theta}}(x) \triangleq \phi^{\boldsymbol{\theta}}(x, t = 1) \tag{1}$$

such that $\phi^{\boldsymbol{\theta}}(x, t = 1)$ solves the integral equation:

$$\phi^{\boldsymbol{\theta}}(x, t) = x + \int_0^t v^{\boldsymbol{\theta}}\big(\phi^{\boldsymbol{\theta}}(x, \tau)\big) \, d\tau. \tag{2}$$

In arbitrary dimensions, computing Definition 3.3 required using an ordinary differential equation solver and can be expensive. However, for 1D diffeomorphisms, as in our DIGRAF, there are closed-form solutions to the CPAB diffeomorphism and its gradients [25], offering an efficient framework for learning activation functions.

### 3.2 Graph Learning Notations

Consider a graph $G = (V, E)$ with $N \in \mathbb{N}$ nodes, where $V = \{1, \ldots, N\}$ is the set of nodes and $E \subseteq V \times V$ is the set of edges. Let $\mathbf{A} \in \{0, 1\}^{N \times N}$ be the adjacency matrix of $G$, and $\mathbf{X} \in \mathbb{R}^{N \times F}$ the node feature matrix,

where $F$ is the number of input features. We denote the feature vector of node $v \in V$ as $\mathbf{x}_v \in \mathbb{R}^F$, which corresponds to the $v$-th row of $\mathbf{X}$. The input node features $\mathbf{X}$ are transformed into the initial node representations $\mathbf{H}^{(0)} \in \mathbb{R}^{N \times C}$, using an embedding function $\mathrm{emb} : \mathbb{R}^F \rightarrow \mathbb{R}^C$ to $\mathbf{X}$, where $C$ is the hidden dimension, that is

$$\mathbf{H}^{(0)} = \mathrm{emb}(\mathbf{X}). \tag{3}$$

The initial features $\mathbf{H}^{(0)}$ are fed to a GNN comprised of $L \in \mathbb{N}$ layers, where each layer $l \in \{1, \ldots, L\}$ is followed by an activation function $\sigma^{(l)}(\cdot; \boldsymbol{\theta}^{(l)}) : \mathbb{R} \rightarrow \mathbb{R}$, and $\boldsymbol{\theta}^{(l)}$ is a set of possibly learnable parameters of $\sigma^{(l)}$. Specifically, the intermediate output of the $l$-th GNN layer is denoted as:

$$\bar{\mathbf{H}}^{(l)} = \mathrm{GNN}^{(l)}_{\mathrm{LAYER}}(\mathbf{H}^{(l-1)}, \mathbf{A}) \tag{4}$$

where $\bar{\mathbf{H}}^{(l)} \in \mathbb{R}^{N \times C}$. The activation function $\sigma^{(l)}$ is then applied *element-wise* to $\bar{\mathbf{H}}^{(l)}$, yielding node features $h_{u,c}^{(l)} = \sigma^{(l)}(\bar{h}_{u,c}^{(l)}; \boldsymbol{\theta}^{(l)}) \, \forall u \in V, \forall c \in [C]$ . Therefore, the application of $\sigma^{(l)}$ can be equivalently written as:

$$\mathbf{H}^{(l)} = \sigma^{(l)}(\bar{\mathbf{H}}^{(l)}; \boldsymbol{\theta}^{(l)}). \tag{5}$$

In the following section, we will show how this abstraction is translated to our DIGRAF.

## 4 DIGRAF

In this section, we formalize our approach, DIGRAF, illustrated in Figure 1, which leverages diffeomorphisms to learn adaptive and flexible graph activation functions.

### 4.1 A CPAB Blueprint for Graph Activation Functions

Our approach builds on the highly flexible CPAB framework [24, 25] and extends it by incorporating Graph Neural Networks (GNNs) to enable the learning of adaptive graph activation functions. While the original CPAB framework was designed for grid deformation and alignment tasks, typically in 1D, 2D, or 3D spaces, we propose a novel application of CPAB in the context of learning activation functions, as described below.

In DIGRAF, we treat a node feature (single channel) as a one-dimensional (1D) point. Given the node features matrix $\bar{\mathbf{H}} \in \mathbb{R}^{N \times C}$, we apply DIGRAF per entry in $\bar{\mathbf{H}}$, in accordance with the typical element-wise computation of activation functions. We mention that, while CPAB was originally designed to learn grid deformations, it can be utilized as an activation function blueprint by considering a conceptual shift that we demonstrate in Figure 4. Given an input function (shown in red in the figure), CPAB deforms grid coordinates, i.e., it transforms it along the horizontal axis, as shown in the blue curve. In contrast, DIGRAF transforms the original data points along the vertical axis, resulting in the green curve. This conceptual shift can be seen visually from the arrows showing the different dimensions of transformations. We therefore refer to the vertical transformation of the data as their activations. Formally, we define the transformation function $T^{(l)}$ as the element-wise application of the diffeomorphism $f^{\boldsymbol{\theta}}$ from Equation (1):

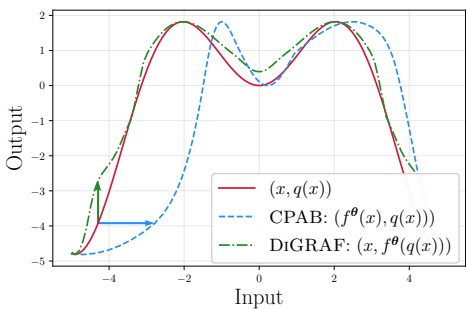

Figure 4: Different transformation strategies. The input function (red), CPAB transformation (blue), and DIGRAF transformation (green), within $\Omega = [-5, 5]$ using the same $\boldsymbol{\theta}$. While CPAB stretches the input, DIGRAF stretches the output, showcasing the distinctive impact of each approach.

$$T^{(l)}(\bar{h}_{u,c}^{(l)}; \boldsymbol{\theta}^{(l)}) \triangleq f^{\boldsymbol{\theta}^{(l)}}(\bar{h}_{u,c}^{(l)}), \tag{6}$$

where $\boldsymbol{\theta}^{(l)}$ denotes learnable parameters of the transformation function $T^{(l)}$, that parameterize the underlying CPA velocity field as discussed in Section 3. In Section 4.2 , we discuss the learning of $\boldsymbol{\theta}^{(l)}$ in DIGRAF.

The transformation $T^{(l)} : \Omega \rightarrow \Omega$ described in Equation (6) is based on CPAB and therefore takes as input values within a domain $\Omega = [a, b]$, and outputs a value within that domain, where $a < b$ are hyperparameters. In practice, we take $a = -b$, such that the activation function can be symmetric and centered around 0, a property known to be desirable for activation functions [16]. For any entry in the intermediate node features $\bar{\mathbf{H}}^{(l)}$(Equation (4)) that is outside the domain $\Omega$, we use the identity function. Therefore, a DIGRAF activation function reads:

$$\mathrm{DIGRAF}(\bar{h}_{u,c}^{(l)}, \boldsymbol{\theta}^{(l)}) = \begin{cases} T^{(l)}(\bar{h}_{u,c}^{(l)}; \boldsymbol{\theta}^{(l)}), & \text{If } \bar{h}_{u,c}^{(l)} \in \Omega \\ \bar{h}_{u,c}^{(l)}, & \text{Otherwise} \end{cases} \tag{7}$$

In practice, DIGRAF is applied element-wise in parallel over all entries, and we use the following notation, which yields the output features post the activation of the $l$-th GNN layer:

$$\mathbf{H}^{(l)} = \text{DIGRAF}(\bar{\mathbf{H}}^{(l)}, \boldsymbol{\theta}^{(l)}). \tag{8}$$

## 4.2 Learning Diffeomorphic Velocity Fields

DIGRAF, defined in Equation (7), introduces graph-adaptivity into the transformation function $T^{(l)}$ by employing an additional GNN, denoted as $\text{GNN}_{\text{ACT}}$, that returns the diffeomorphism parameters $\boldsymbol{\theta}^{(l)}$:

$$\boldsymbol{\theta}^{(l)}(\bar{\mathbf{H}}^{(l)}, \mathbf{A}) = \text{POOL}\left(\text{GNN}_{\text{ACT}}(\bar{\mathbf{H}}^{(l)}, \mathbf{A})\right), \tag{9}$$

where POOL is a graph-wise pooling operation, such as max or mean pooling. The resulting vector $\boldsymbol{\theta}^{(l)} \in \mathbb{R}^{\mathcal{N}_{\mathcal{P}}-1}$, which is dependent on the tessellation size $\mathcal{N}_{\mathcal{P}}$, is then used to compute the output of the $l$-th layer, $\mathbf{H}^{(l)}$, as described in Equation (8). We note that Equation (9) yields a different $\boldsymbol{\theta}^{(l)}$ for every input graph and features pair $(\bar{\mathbf{H}}^{(l)}, \mathbf{A})$, which implies the graph-adaptivity of DIGRAF. Furthermore, since $\text{GNN}_{\text{ACT}}$ is trained with the other network parameters in an end-to-end fashion, DIGRAF is also adaptive to the task of interest. In Appendix B, we provide and discuss the implementation details of $\text{GNN}_{\text{ACT}}$ and POOL.

**Variants of DIGRAF.** Equation (9) describes an approach to introduce graph-adaptivity to $\boldsymbol{\theta}^{(l)}$ using $\text{GNN}_{\text{ACT}}$. An alternative approach is to directly optimize the parameters $\boldsymbol{\theta}^{(l)} \in \mathbb{R}^{\mathcal{N}_{\mathcal{P}}-1}$, without using an additional GNN. Note that in this case, input and graph-adaptivity are sacrificed in favor of a computationally lighter solution. We denote this variant of our method by DIGRAF (W/O ADAP.). Considering this variant is important because it allows us to: (i) offer a middle-ground solution in terms of computational effort, and (ii) it allows us to directly quantify the contribution of graph-adaptivity in DIGRAF. In Section 5, we compare the performance of the methods.

**Velocity Field Regularization.** To ensure the smoothness of the velocity field, which will encourage training stability [81], we incorporate a regularization term in the learning procedure of $\boldsymbol{\theta}^{(l)}$. Namely, we follow the Gaussian smoothness prior on the CPA velocity field from Freifeld et al. [24], which was shown to be effective in maintaining smooth transformations. The regularization term is defined as follows:

$$\mathcal{R}(\{\boldsymbol{\theta}^{(l)}\}_{l=1}^{L}) = \sum_{l=1}^{L} \boldsymbol{\theta}^{(l)\top} \Sigma_{\text{CPA}}^{-1} \boldsymbol{\theta}^{(l)}, \tag{10}$$

where $\Sigma_{\text{CPA}}$ represents the covariance of a zero-mean Gaussian smoothness prior defined as in Freifeld et al. [24]. We further maintain the boundedness of $\boldsymbol{\theta}^{(l)}$ by employing a hyperbolic tangent function (Tanh). In this way, $\boldsymbol{\theta}^{(l)}$ remains in $[-1, 1]$ when applied in $T^{(l)}$ in Equation (7), ensuring that the velocity field parameters remain bounded, encouraging the overall training stability of the model.

## 4.3 Properties of DIGRAF

In this section, we focus on understanding the theoretical properties of DIGRAF, highlighting the compelling attributes that establish it as a performant activation function for GNNs.

**DIGRAF yields differentiable activations.** By construction, DIGRAF learns a diffeomorphism, which is differentiable by definition. Being differentiable everywhere is considered beneficial as it allows for smooth weight updates during backpropagation, preventing the zigzagging effect in the optimization process [77].

**DIGRAF is bounded within the input-output domain $\Omega$.** We point out in Remark D.3 that the diffeomorphism $T^{(l)}(\cdot; \boldsymbol{\theta}^{(l)})$ is a $\Omega \to \Omega$ transformation. Any diffeomorphism is continuous, and by the extreme value theorem, $T^{(l)}(\cdot; \boldsymbol{\theta}^{(l)})$ is bounded in $\Omega$. This prevents the activation values from becoming excessively large, a property linked to faster convergence [16].

**DIGRAF can learn to be zero-centered.** Benefiting from its flexibility, DIGRAF has the capacity to learn activation functions that are inherently zero-centered. As an input-adaptive activation function governed by a parameters vector $\boldsymbol{\theta}^{(l)}$, DIGRAF can be adjusted through $\boldsymbol{\theta}^{(l)}$ to maintain a zero-centered nature. This property is associated with accelerated convergence in neural network training [16].

**DIGRAF is efficient.** DIGRAF exhibits linear computational complexity, and can further achieve sub-linear running times via parallelization in practice [25]. Moreover, with the existence of a closed-form solution for $f^{\boldsymbol{\theta}^{(l)}}$ and its gradient in the 1D case [24], the computations of CPAB can be done efficiently. Additionally, the measured runtimes, detailed in Appendix H, underscore the complexity comparability of DIGRAF with other graph activation functions.

In addition to the above properties, which follow from our design choice of learning diffeomorphisms through the CPAB framework, we briefly present the following properties, which are formalized and proven in Appendix D.

**DIGRAF is permutation equivariant.** We demonstrate in Proposition D.4 that DIGRAF exhibits permutation equivariance to node numbering, ensuring that its behavior remains consistent regardless of the ordering of the graph nodes, which is a key desired property in designing GNN components [8].

**DIGRAF is Lipschitz continuous.** We show in Proposition D.2 that DIGRAF is Lipschitz continuous and derive its Lipschitz constant. Since it is also bounded, we can combine the two results, which leads us to the following proposition:

**Proposition 4.1** (The boundedness of $T(\cdot; \boldsymbol{\theta}^{(l)})$ in DIGRAF). *Given a bounded domain* $\Omega = [a, b] \subset \mathbb{R}$ *where* $a < b$, *and any two arbitrary points* $x, y \in \Omega$, *the maximal difference of a diffeomorphism* $T(\cdot; \boldsymbol{\theta}^{(l)})$ *with parameter* $\boldsymbol{\theta}^{(l)}$ *in* DIGRAF *is bounded as follows:*

$$|T(x; \boldsymbol{\theta}^{(l)}) - T(y; \boldsymbol{\theta}^{(l)})| \leq \min(|b - a|, |x - y| \exp(C_{v^{\boldsymbol{\theta}^{(l)}}})) \tag{11}$$

*where* $C_{v^{\boldsymbol{\theta}^{(l)}}}$ *is the Lipschitz constant of the CPA velocity field* $v^{\boldsymbol{\theta}^{(l)}}$.

**DIGRAF extends commonly used activation functions.** CPAB [24, 25], which is used as a framework to learn the diffeomorphism in DIGRAF, is capable of learning and representing a wide range of diffeomorphic functions. When used as an activation function, the transformation $T^{(l)}(\cdot; \boldsymbol{\theta}^{(l)})$ in DIGRAF adapts to the specific graph and task by learning different $\boldsymbol{\theta}^{(l)}$ parameters, rather than having a fixed diffeomorphism. Examples of popular and commonly used diffeomorphisms utilized as activations include Sigmoid, Tanh, Softplus, and ELU, as we show in Appendix D. Extending this approach is our DIGRAF that learns the diffeomorphism during training rather than selecting a pre-defined function.

## 5 Experiments

In this section, we conduct an extensive set of experiments to demonstrate the effectiveness of DIGRAF as a graph activation function. Our experiments seek to address the following questions:

(Q1) Does DIGRAF consistently improve the performance of GNNs compared to existing activation functions on a broad set of downstream tasks?

(Q2) To what extent is graph-adaptivity in DIGRAF beneficial when compared to our baseline of DIGRAF (W/O ADAP.) and existing activation functions that lack adaptivity?

(Q3) Compared with other graph-adaptive activation functions, how does the added flexibility offered by DIGRAF impact downstream performance?

(Q4) How do the considered activation functions compare in terms of training convergence?

**Baselines.** We compare DIGRAF with three categories of relevant and competitive baselines: (1) *Standard Activation Functions*, namely Identity, Sigmoid [69], ReLU [26], LeakyReLU [50], Tanh [35], GeLU [34], and ELU [12] to estimate the benefit of learning activation functions parameters; (2) *Learnable Activation Functions*, specifically PReLU [33], Maxout [29] and Swish [66], to assess the value of graph-adaptivity; and (3) *Graph Activation Functions*, such as Max [38], Median [38] and GReLU [92], to evaluate the effectiveness of DIGRAF's design in capturing graph structure and the blueprint flexibility of DIGRAF as discussed in Section 4.

All baselines are integrated into GCN [43] for node tasks and GIN [84] (GINE [37] where edge features are available) for graph tasks, to ensure fair and meaningful comparisons, isolating the impact of other design choices. We provide additional details on the experimental settings and datasets in Appendix G, as well as additional experiments, including ablation studies, in Appendix E.

### 5.1 Node Classification

Our results are summarized in Table 1, where we consider the BLOGCATALOG [86], FLICKR [86], CITESEER [73], CORA [53], and PUBMED [58] datasets. As can be seen from the Table, DIGRAF consistently outperforms all standard activation functions, as well as all the learnable activation functions. Additionally, DIGRAF outperforms other graph-adaptive activation functions. We attribute this positive performance gap to the ability of DIGRAF to learn complex non-linearities due to its diffeomorphism-based blueprint, compared to piecewise linear or pre-defined functions as in other methods. Finally, we compare the performance of DIGRAF and DIGRAF (W/O ADAP.). We remark that in this experiment, we are operating in a transductive setting, as the data consists of a single graph, implying that both DIGRAF and DIGRAF (W/O ADAP.) are adaptive in this case. Still, we see that DIGRAF slightly outperforms the DIGRAF (W/O ADAP.) and we attribute this

Table 1: Comparison of node classification accuracy (%) ↑ on different datasets using various baselines with DIGRAF. The top three methods are marked by **First**, **Second**, **Third**.

| Method ↓ / Dataset → | BLOG CATALOG | FLICKR | CITESEER | CORA | PUBMED |
|---|---|---|---|---|---|
| **STANDARD ACTIVATIONS** | | | | | |
| GCN + Identity | 74.8±0.5 | 53.5±1.1 | **69.1±1.6** | 80.5±1.2 | 77.6±2.1 |
| GCN + Sigmoid [69] | 39.7±4.5 | 18.3±1.2 | 27.9±2.1 | 32.1±2.3 | 52.8±6.6 |
| GCN + ReLU [43] | 72.1±1.9 | 50.7±2.3 | 67.7±2.3 | 79.2±1.4 | 77.6±2.2 |
| GCN + LeakyReLU [50] | 72.6±2.1 | 51.0±2.0 | 68.4±1.8 | 79.4±1.6 | 76.8±1.6 |
| GCN + Tanh [35] | 73.9±0.5 | 51.3±1.5 | **69.1±1.4** | 80.5±1.3 | **77.9±2.1** |
| GCN + GeLU [34] | 75.8±0.5 | **56.1±1.3** | 67.8±1.7 | 79.3±1.9 | 77.1±2.7 |
| GCN + ELU [12] | 74.8±0.5 | 53.4±1.1 | **69.1±1.7** | **80.7±1.2** | 77.5±2.2 |
| **LEARNABLE ACTIVATIONS** | | | | | |
| GCN + PReLU [33] | 74.8±0.4 | 53.2±1.5 | **69.2±1.5** | 80.5±1.2 | 77.6±2.1 |
| GCN + Maxout [29] | 72.4±1.4 | 54.0±1.8 | 68.5±2.2 | 79.8±1.5 | 77.3±2.9 |
| GCN + Swish [66] | **76.0±0.7** | 55.7±1.4 | 67.7±1.8 | 79.2±1.1 | 77.3±2.8 |
| **GRAPH ACTIVATIONS** | | | | | |
| GCN + Max [38] | 72.0±1.0 | 47.5±0.9 | 59.7±2.9 | 76.0±1.8 | 75.0±1.4 |
| GCN + Median [38] | **77.7±0.7** | **58.3±0.6** | 61.3±2.7 | 77.1±1.1 | 75.7±2.5 |
| GCN + GReLU [92] | 73.7±1.2 | 54.4±1.6 | 68.5±1.9 | **81.8±1.8** | **78.9±1.7** |
| GCN + DIGRAF (W/O ADAP.) | 80.8±0.6 | 68.6±1.8 | 69.2±2.1 | 81.5±1.1 | 78.3±1.6 |
| GCN + DIGRAF | **81.6±0.8** | **69.6±0.6** | **69.5±1.4** | **82.8±1.1** | **79.3±1.4** |

performance gain to the GNN layers within DIGRAF that are (i) explicitly graph-aware, and (ii) can facilitate the learning of better diffeomorphism parameters $\boldsymbol{\theta}^{(l)}$ (Equation (9)) due to the added complexity.

## 5.2 Graph Classification and Regression

**ZINC-12K**. In Table 2 we present results on the ZINC-12K [75, 31, 18] dataset for the regression of constrained solubility of molecules. We note that DIGRAF achieves an MAE of 0.1302, surpassing the best-performing activation on this dataset, Maxout, by 0.0285, which translates to a relative improvement of ~ 18%.

**OGB.** We evaluate DIGRAF on 4 datasets from the OGB benchmark [36], namely, MOLESOL, MOLTOX21, MOLBACE, and MOL-HIV. The results are reported in Table 3, where it is noted that DIGRAF achieves significant improvements compared to standard, learnable, and graph-adaptive activation functions. For instance, DIGRAF obtains a ROC-AUC score of 80.28% on MOLHIV, an absolute improvement of 4.7% over the best performing activation function (ReLU).

**TUDatasets..** In addition to the aforementioned datasets, we evaluate DIGRAF on the popular TUDatasets [56]. We present results on MUTAG, PTC, PROTEINS, NCI1 and NCI109 in Table 5 in Appendix E. The results show that DIGRAF is always within the top-three performing activations across all datasets. As an example, on PROTEINS dataset, we see an absolute improvement of 1.1% over the best-performing activation functions (Maxout and GReLU).

Table 2: Comparison on ZINC-12K under the 500K parameter budget. The top three methods are **First**, **Second**, **Third**.

| Method | ZINC (MAE ↓) |
|---|---|
| **STANDARD ACTIVATIONS** | |
| GIN + Identity | 0.2460±0.0214 |
| GIN + Sigmoid [69] | 0.3839±0.0058 |
| GIN + ReLU [84] | **0.1630±0.0040** |
| GIN + LeakyReLU [50] | 0.1718±0.0042 |
| GIN + Tanh [35] | 0.1797±0.0064 |
| GIN + GeLU [34] | 0.1896±0.0023 |
| GIN + ELU [12] | 0.1741±0.0089 |
| **LEARNABLE ACTIVATIONS** | |
| GIN + PReLU [33] | 0.1798 ±0.0067 |
| GIN + Maxout [29] | **0.1587±0.0057** |
| GIN + Swish [66] | 0.1636±0.0039 |
| **GRAPH ACTIVATIONS** | |
| GIN + Max [38] | 0.1661±0.0035 |
| GIN + Median [38] | 0.1715±0.0050 |
| GIN + GReLU [92] | 0.3003±0.0086 |
| GIN + DIGRAF (W/O ADAP.) | 0.1382±0.0080 |
| GIN + DIGRAF | **0.1302±0.0090** |

## 5.3 Convergence Analysis

Besides improved downstream performance, another important aspect of activation functions is their contribution to training convergence [16]. We therefore present the training curves of DIGRAF as well as the rest of the considered baselines to gain insights into their training convergence. Results for representative datasets are presented in Figure 5, where DIGRAF achieves similar or better training convergence than other methods, while also demonstrating better generalization abilities due to its better performance.

## 5.4 Discussion

Our extensive experiments span across 15 different datasets and benchmarks, consisting of both node- and graph-level tasks. Our key takeaways are as follows:

Table 3: A comparison of DIGRAF to natural baselines, standard, and graph activation layers on OGB datasets, demonstrating the advantage of our approach. The top three methods are marked by **First**, **Second**, **Third**.

| Method ↓ / Dataset → | MOLESOL RMSE ↓ | MOLTOX21 ROC-AUC ↑ | MOLBACE ROC-AUC ↑ | MOLHIV ROC-AUC ↑ |
|---|---|---|---|---|
| **STANDARD ACTIVATIONS** | | | | |
| GIN + Identity | 1.402±0.036 | 74.51±0.44 | 72.69±2.93 | 75.12±0.77 |
| GIN + Sigmoid [69] | 0.884±0.043 | 69.15±0.52 | 68.70±3.68 | 73.87±0.80 |
| GIN + ReLU [84] | 1.173±0.057 | 74.91±0.51 | 72.97±4.00 | **75.58±1.40** |
| GIN + LeakyReLU [50] | 1.219±0.055 | 74.60±1.10 | 73.40±3.19 | 74.75±1.20 |
| GIN + Tanh [35] | 1.190±0.044 | 74.93±0.61 | 74.92±2.47 | **75.22±2.03** |
| GIN + GeLU [34] | 1.147±0.050 | 74.29±0.59 | 75.59±3.32 | 74.15±0.79 |
| GIN + ELU [12] | 1.104±0.038 | 75.08±0.62 | 76.10±3.29 | 75.09±0.65 |
| **LEARNABLE ACTIVATIONS** | | | | |
| GIN + PReLU [33] | **1.098±0.062** | 74.51±0.92 | 76.16±2.28 | 73.56±1.63 |
| GIN + Maxout [29] | 1.109±0.045 | 75.14±0.87 | 76.83±3.88 | 72.75±2.10 |
| GIN + Swish [66] | 1.113±0.066 | 73.31±1.01 | **77.23±2.35** | 72.95±0.64 |
| **GRAPH ACTIVATIONS** | | | | |
| GIN + Max [38] | 1.199±0.070 | **75.50±0.77** | 77.04±2.81 | 73.44±2.08 |
| GIN + Median [38] | **1.049±0.038** | 74.39±0.90 | **77.26±2.74** | 72.80±2.21 |
| GIN + GReLU [92] | 1.108±0.066 | **75.33±0.51** | 75.17±2.60 | 73.45±1.62 |
| GIN + DIGRAF (W/O ADAP.) | 0.9011±0.047 | 76.37±0.49 | 78.90±1.41 | 79.19±1.36 |
| GIN + DIGRAF | **0.8196±0.051** | **77.03±0.59** | **80.37±1.37** | **80.28±1.44** |

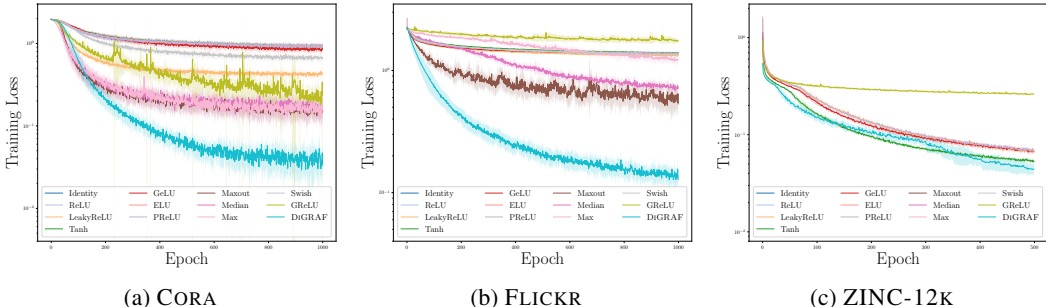

(a) CORA        (b) FLICKR        (c) ZINC-12K

Figure 5: Convergence analysis of DIGRAF compared to baseline activation functions. The plot illustrates the training loss over epochs, showcasing the overall faster convergence of DIGRAF.

(A1) **Overall Performance of DIGRAF:** The performance offered by DIGRAF is consistent and on par with or better than other activation functions, across all datasets. These results establish DIGRAF as a highly effective approach for learning graph activation functions.

(A2) **Benefit of Graph-Adaptivity:** DIGRAF outperforms the learnable (although not graph-adaptive) activation functions such as PReLU, Maxout, and Swish, as well as our non-graph adaptive baseline DIGRAF (W/O ADAP.), on all considered datasets. This observation highlights the crucial role of graph-adaptivity in activation functions for GNNs.

(A3) **The Benefit of Blueprint Flexibility:** DIGRAF consistently outperforms other graph-adaptive activation functions like Max, Median, and GReLU. We tie this positive performance gap to the ability of DIGRAF to model complex non-linearities due to its diffeomorphism-based blueprint, compared to piecewise linear or pre-defined functions as in other methods.

(A4) **Convergence of DIGRAF:** As shown in Section 5.3, in addition to overall better downstream performance, DIGRAF allows to achieve better training convergence.

In summary, compared with 12 well-known activation functions used in GNNs, and across multiple datasets and benchmarks, DIGRAF demonstrates a superior, learnable, flexible, and versatile graph-adaptive activation function, highlighting it as a strong approach for designing and learning graph activation functions.

## 6 Conclusions

In this work, we introduced DIGRAF, a novel activation function designed for graph-structured data. Our approach leverages Continuous Piecewise-Affine Based (CPAB) transformations to integrate a graph-adaptive mechanism, allowing DIGRAF to adapt to the unique structural features of input graphs. We show that DIGRAF

exhibits several desirable properties for an activation function, including differentiability, boundedness within a defined interval, and computational efficiency. Furthermore, we demonstrated that DIGRAF maintains stability under input perturbations and is permutation equivariant, therefore suitable for graph-based applications. Our extensive experiments on diverse datasets and tasks demonstrate that DIGRAF consistently outperforms traditional, learnable, and existing graph-specific activation functions.

**Limitations and Broader Impact.** While DIGRAF demonstrates consistent superior performance compared to existing activation functions, there remain areas for potential improvement. For instance, the current formulation is limited to learning activation functions that belong to the class of diffeomorphisms, which, despite encompassing a wide range of functions, might not be optimal. By improving the performance on real-world tasks like molecule property prediction, and offering faster training convergence, we envision a positive societal impact by DIGRAF in drug discovery and in achieving a lower carbon footprint.

### Acknowledgments

BR acknowledges support from the National Science Foundation (NSF) awards, CCF-1918483, CAREER IIS-1943364 and CNS-2212160, Amazon Research Award, AnalytiXIN, and the Wabash Heartland Innovation Network (WHIN), Ford, NVidia, CISCO, and Amazon. Computing infrastructure was supported in part by CNS-1925001 (CloudBank). This work was supported in part by AMD under the AMD HPC Fund program. ME is funded by the Blavatnik-Cambridge fellowship, the Cambridge Accelerate Programme for Scientific Discovery, and the Maths4DL EPSRC Programme. The authors thank Shahaf Finder, Ron Shapira-Weber, and Oren Freifeld for the discussions on CPAB.

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

# A   Additional Related Work

**Graph Neural Networks.**   Graph Neural Networks [71] (GNNs) have emerged as a transformative approach in machine learning, notably following the popularity of the message-passing scheme [28]. GNNs enable effective learning from graph-structured data, and can be applied to different tasks, ranging from social network analysis [43] to bioinformatics [40]. In recent years, various GNN architectures were proposed, aiming to address various aspects, from alleviating oversmoothing [10], concerning attention mechanisms in the message passing scheme [78, 7, 42], or focusing on the expressive power of the architectures [57, 23, 88, 89, 65, 4], given that message-passing based architectures are known to be bounded by the WL graph isomorphism test [84, 55].

Despite advancements, the poor performance of deep GNNs has led to a preference for shallow architectures GCNs [47]. To enhance performance, techniques such as pooling functions have been proposed, introducing generalization by reducing feature map sizes [93]. Methods such as HGP-SL [93], GraphUNet [27], and LaPool [59] introduce pooling layers specifically designed for GNNs. Beyond node feature, the importance of graph structure and positional features is increasingly recognized, with advancements such as GraphGPS [67] and SAN [44] integrating positional and structural encodings through attention-based mechanisms.

**Evaluation of Rectified Activation Functions.**   Rectified activation functions, represented by the Rectified Linear Unit (ReLU), have been widely applied and studied in various neural network architectures due to their simplicity and effectiveness [60, 3, 45]. The prevalent assumption that ReLU's performance is predominantly due to its sparsity is critically examined by Xu et al. [83], suggesting introducing a non-zero slope in the negative part can significantly enhance network performance. Extending this, Price et al. [64] investigates sparsity-inducing activation functions, such as the shifted ReLU, in network initialization and early stages of training. These functions can mitigate overfitting and boost model generalization capabilities. Conversely, it was shown that in overparameterized networks, smoother activation functions, like Tanh and Swish, can enhance the convergence rate, in contrast to the non-smooth characteristics of ReLU [62]. However, the fixed nature of ReLU and many of its variants restricts their ability to adapt the input, resulting in limited power to capture dynamics in learning.

**Advancements in Learnable Activation Functions.**   Recent research has increasingly focused on adaptive and learnable activation functions, which are optimized alongside the learning process of the network. The AdAct framework [51] introduces learnability by combining multiple activation functions into a single module with learnable weighting coefficients. However, these coefficients are fixed after training, limiting the framework's adaptability to varying inputs. A concurrent work by Liu et al. [49] introduces Kolmogorov-Arnold Networks (KAN), a novel architecture that diverges from traditional Multi-Layer Perceptron (MLP) configurations, which applies activation functions to network edges instead of nodes. Unlike our current work, which focuses only on the design of activation functions for GNNs, their research extends beyond this scope and considers a fundamental architecture design. Finally, the recently proposed TAFS [85] learns a task-adaptive (but not graph-adaptive) activation function for GNNs through a bi-level optimization.

# B   Implementation Details of DıGRAF

**Multiple Graphs in one Batch.**   Consider a set of graphs $S = \{G_1, G_2, \cdots, G_B\}$ with a batch size of $B$. Let $N_S = N_1 + N_2 + \cdots + N_B$ represent the cumulative number of nodes across the graph dataset. The term $N_{\max} \triangleq \max(N_1, N_2, \cdots, N_B)$ denotes the largest node count present in any single graph within $S$.

To create a unified feature matrix for $S$ that encompasses all graphs in the batch, we standardize the dimension by padding each feature matrix $\mathbf{X}_i \in \mathbb{R}^{N_i \times C}$, $i \in [B]$ for graph $G_i \in S$ from $N_i$ to $N_{\max}$ with zeros. The combined feature matrix $\mathbf{X}_S$ is constructed by concatenating the transposed feature matrices $\mathbf{X}_i^\top \; \forall i \in [B]$, resulting in a matrix that lies in the domain $\mathbb{R}^{(B \cdot C) \times N_{\max}}$. This matrix is permutation invariant; while relabeling nodes changes the row indices, it does not affect the overall transformation process. Therefore, DıGRAF can handle multiple graphs in a batch. In practice, to avoid the overhead of padding, we use the batching support from Pytorch-Geometric [22].

**Implementation Details of $\text{GNN}_{\text{ACT}}$.**   In Section 4.2, we examined two distinct approaches to learn the diffeomorphism parameters $\boldsymbol{\theta}^{(l)}$, either directly or through $\text{GNN}_{\text{ACT}}$. As shown in Appendix C, $\boldsymbol{\theta}^{(l)}$ determines the velocity field $v^{\boldsymbol{\theta}^{(l)}}$. Predicting a graph-dependent $\boldsymbol{\theta}^{(l)}$ adds graph-adaptivity to the activation function $T^{(l)}$. In DıGRAF we achieve this by employing another GNN $\text{GNN}_{\text{ACT}}$, described below.

The backbone of $\text{GNN}_{\text{ACT}}$ utilizes the same structure as the primary network layers $\text{GNN}_{\text{LAYER}}^{(l)}$, that is, GCN [43] or GIN [84]. It is important to note, that while $\text{GNN}_{\text{ACT}}$ has a similar structure to the primary network GNN with ReLU activation function, it has its own set of learnable weights, and it is shared among the layers, unlike the primary GNN layers $\text{GNN}_{\text{LAYER}}^{(l)}$. The hidden dimensions and the number of layers of $\text{GNN}_{\text{ACT}}$ are

hyperparameters. The weight parameters of GNN_ACT are trained concurrently with the main network weights. As described in Equation (9), after the computation of GNN_ACT, a pooling layer denoted by POOL is placed to aggregate node features. This aggregation squashes the node dimension such that the output is not dependent on the specific order of nodes, and it yields the vector of parameters $\boldsymbol{\theta}^{(l)}$.

**Rescaling $\bar{\mathbf{H}}^{(l)}$.** Following the implementation of Freifeld et al. [24], the default 1D domain for CPAB is set as $[0, 1]$. To enhance the flexibility of $T^{(l)}$ and ensure its adaptability across various input datasets, DIGRAF extends the domain to $\Omega = [a, b] \subset \mathbb{R}$ with $a < b$ as shown in Section 3.1. To match the two domains, we rescale the intermediate feature matrix $\bar{\mathbf{H}}^{(l)}$ from $\Omega$ to the unit interval $[0, 1]$ before passing it to $T^{(l)}$. Let $r = \frac{b-a}{2}$, then rescaling is performed using the function $f(x) = (x + r)/(2r)$. Data points outside this range will retain their original value, effectively acting as an identity function outside the domain $\Omega$.

**Training Loss Function.** As described in Equation (10), we employ a regularization term for the velocity field to maintain the smoothness of the activation function. To control the strength of regularization, we introduce a hyperparameter $\lambda$. We denote $\mathcal{L}_{\text{TASK}}$ as the loss function of the downstream task (i.e. cross-entropy loss in case of classification and mean absolute error in case of regression tasks), and the overall training loss of DIGRAF, denoted as $\mathcal{L}_{\text{TOTAL}}$ is given as

$$\mathcal{L}_{\text{TOTAL}} = \mathcal{L}_{\text{TASK}} + \lambda \, \mathcal{R}(\{\boldsymbol{\theta}^{(l)}\}_{l=1}^{L}). \tag{12}$$

## C  Overview of CPA Velocity Fields and CPAB Transformations

In this Section, we drop the layer notations $l$ for simplicity. In Section 3.1, we introduce the concept of a diffeomorphism on a closed interval in Definition 3.1, which can be learned through the integration of a Continuous Piecewise Affine (CPA) velocity field. As detailed in Definition 3.2, the velocity field $v^{\boldsymbol{\theta}}$ is governed by the parameter $\boldsymbol{\theta}$ and the tessellation $\mathcal{P}$. We now discuss how the velocity fields are computed following the methodologies presented by Freifeld et al. [24, 25] and highlight the relations between $v^{\boldsymbol{\theta}}$, $\boldsymbol{\theta}$ and $\mathcal{P}$. We start by formally defining the tessellation on $\Omega$:

**Definition C.1** (Tessellation of a closed interval [24]). A tessellation $\mathcal{P}$ of size $\mathcal{N}_{\mathcal{P}}$ subintervals of a closed interval $\Omega = [a, b]$ in $\mathbb{R}$ is a partitioning $\{[x_i, x_{i+1}]\}_{i=0}^{\mathcal{N}_{\mathcal{P}}-1}$ that satisfies the following properties:

(1)  $x_0 = a$ and $x_{\mathcal{N}_{\mathcal{P}}} = b$

(2)  Each point $x \in \Omega$ lies in at least one subinterval $[x_i, x_{i+1}]$

(3)  The intersection of any two subintervals $[x_i, x_{i+1}]$ and $[x_{i+1}, x_{i+2}]$ is exactly $\{x_{i+1}\}$

(4)  $\bigcup_{i=0}^{\mathcal{N}_{\mathcal{P}}-1} [x_i, x_{i+1}] = \Omega$

The vector of parameters $\boldsymbol{\theta}$ is linked to the subintervals in $\mathcal{P}$, whose dimension is determined by the number of intervals $\mathcal{N}_{\mathcal{P}}$. Similar to Freifeld et al. [24], we impose boundary constraints that mandate the velocity at the boundary of the tessellation to be zero, i.e., $v^{\boldsymbol{\theta}}(0) = v^{\boldsymbol{\theta}}(1) = 0$. This boundary condition allows us to compose the diffeomorphism in the domain $\Omega$ with an identity function for any values outside the domain. Under this constraint, the degrees of freedom (number of parameters) for $\theta$ is $\mathcal{N}_{\mathcal{P}} - 1$.

The velocity field is then defined as follows:

**Definition C.2** (Relation between $\boldsymbol{\theta}$ and $v^{\boldsymbol{\theta}}$, taken from Freifeld et al. [25]). Given a tessellation $\mathcal{P}$ with $\mathcal{N}_{\mathcal{P}}$ intervals on a closed domain $\Omega = [a, b]$, as defined in Definition C.1. Given a parameter $\boldsymbol{\theta} \in \mathbb{R}^{\mathcal{N}_{\mathcal{P}}-1}$ and an arbitrary point $x$ within the domain, a continuous piecewise-affine velocity field $v^{\boldsymbol{\theta}}$ can present as follows:

$$v^{\boldsymbol{\theta}}(x) = \sum_{j=0}^{\mathcal{N}_{\mathcal{P}}-2} \boldsymbol{\theta}_j \mathbf{b}_j \tilde{x}, \tag{13}$$

where $\{\mathbf{b}_j\}_{j=0}^{\mathcal{N}_{\mathcal{P}}-2}$ is an orthonormal basis of the space of velocity fields $\mathcal{V}$, such that $v^{\boldsymbol{\theta}} \in \mathcal{V}$, and $\tilde{\boldsymbol{x}} = \begin{bmatrix} x \\ 1 \end{bmatrix}$.

The orthonormal basis $\{\mathbf{b}_j\}_{j=0}^{\mathcal{N}_{\mathcal{P}}-2}$ for the velocity field can be obtained through Singular Value Decomposition of $\mathbf{L}$. Note that $\mathbf{L}$ is a matrix constraining the coefficients of each continuous piecewise-affine velocity function by ensuring that the velocity value at the shared endpoints is the same [52]. Let $vec(\mathbf{A})$ be a column vector containing the coefficients for each interval, for instance, $[a_0, b_0, a_1, b_1]^T$ for consecutive intervals interval 0 and interval 1. The shared endpoint is $x_1$. To achieve the constrain, we have the equation $a_0 * x_1 + b_0 + a_1 * (-x_1) + b_1 * (-1) = 0$. In this example, the constrain matrix $\mathbf{L}$ is $\mathbf{L} = [x_1, 1, -x_1, -1]$.

Table 4: A summary of the properties of activation functions. – means not studied in the corresponding paper. * Median-of-medians algorithm can achieve linear time complexity on average.

| Act ↓ / Prop → | Boundedness | Differentiability | Linear Complexity | Permutation Equiv. | Lipschitz Cont. | Graph Adap. |
|---|---|---|---|---|---|---|
| ReLU [84] | ✗ | ✗ | ✓ | ✓ | ✓ | ✗ |
| Tanh [35] | ✓ | ✓ | ✓ | ✓ | ✓ | ✗ |
| PReLU [33] | ✗ | ✗ | ✓ | ✓ | ✓ | ✗ |
| Swish [66] | ✗ | ✓ | ✓ | ✓ | ✓ | ✗ |
| Max [38] | ✗ | ✗ | ✓ | ✓ | ✓ | ✓ |
| Median [38] | ✗ | ✗ | ✓* | ✓ | – | ✓ |
| GReLU [92] | ✗ | ✗ | ✓ | ✓ | – | ✓ |
| DIGRAF | ✓ | ✓ | ✓ | ✓ | ✓ | ✓ |

By generalizing the previous example, the constraint can be expressed as $\mathbf{L} * vec(\mathbf{A}) = \vec{0}$. As the endpoints are decided by the tessellation setup, we can build the constrain matrix $\mathbf{L}$ without knowing $vec(\mathbf{A})$. And thus, orthonormal basis $\{\mathbf{b}_j\}_{j=0}^{\mathcal{N}_\mathcal{P}-2}$ can be computed by giving tessellation setup.

**Proposition C.3** (DIGRAF has a closed form solution). *Equation 2 can be expressed as an equivalent ODE. By allowing $x$ to vary and fixing $t$, the solution to this ODE can be written as a composition of a finite number of solutions $\psi$:*

$$\phi^\theta(x,t) = (\psi_{\theta,c_m}^{t_m} \circ \psi_{\theta,c_{m-1}}^{t_{m-1}} \circ \cdots \circ \psi_{\theta,c_2}^{t_2} \circ \psi_{\theta,c_1}^{t_1})(x)$$

*Here $m$ represents the number of cells visited. Given $x, \theta$, time $t$, and the smallest cell index containing $x$, $c$, we can compute each $\psi_{\theta,c_i}^{t_i}(x), i \in \{1, \ldots, m\}$ from $\psi_{\theta,c_1}^{t_1}(x)$ to $\psi_{\theta,c_m}^{t_m}(x)$. In other words, DIGRAF has a closed-form solution.*

*Proof.* The proof follows the steps in Martinez et al. [52]. Equation (2) can be expressed as the equivalent ODE: $\frac{d\phi^\theta(x,t)}{dt} = v^\theta(\phi^\theta(x,t))$. By allowing $x$ to vary and fixing $t$, the solution to this ODE can be written as a composition of a finite number of solutions $\psi$:

$$\phi^\theta(x,t) = (\psi_{\theta,c_m}^{t_m} \circ \psi_{\theta,c_{m-1}}^{t_{m-1}} \circ \cdots \circ \psi_{\theta,c_2}^{t_2} \circ \psi_{\theta,c_1}^{t_1})(x)$$

Given $x, \theta$, time $t$, and a function $\gamma$ that returns the smallest cell index containing $x$, namely $c = \gamma(x)$, then we can compute each $\psi_{\theta,c_i}^{t_i}(x), i \in \{1, \ldots, m\}$ and use them in the closed form solution for the ODE. The cell boundary $x_c$ is determined based on the velocity value $v(x)$ at point $x$. If $v(x) \geq 0$, $x_c$ is the largest point in the interval; otherwise, it is the smallest point. In this setup, a cell is a 1D interval with two endpoints. At the hitting time $t_{hit}$, $\psi_c^\theta(x, t_{hit})$ is

$$\psi_c^\theta(x, t_{hit}) = x_c,$$

where $t_{hit}^\theta = \frac{1}{a_c^\theta} \log \left( \frac{a_c^\theta x_c + b_c^\theta}{a_c^\theta x + b_c^\theta} \right)$. The CPAB velocity field is continuous piecewise-affine, and for each interval with index $c$, it has coefficients $a_c^\theta$ (slope) and $b_c^\theta$ (bias). These can be computed given $\theta$. If $t_{hit}^\theta > t$, then $\phi^\theta(x,t) = \psi_c(x,t)$. Otherwise, we repeat the process with updated values $t = t - t_{hit}^\theta$, $x = x_c$, and $c$ adjusted based on the sign of $v(x)$.

This iterative process continues until convergence, with an upper bound for $m$ being $max(c_1, N_P - c_1 + 1)$, where $c_1$ refers to the first visited cell index, and $N_P$ is the number of closed intervals in the space $\Omega$. With the above steps, we can precisely compute each $\psi_{\theta,c_i}^{t_i}(x), i \in \{1, \ldots, m\}$ from $\psi_{\theta,c_1}^{t_1}(x)$ to $\psi_{\theta,c_m}^{t_m}(x)$ following the equation $\phi^\theta(x,t) = (\psi_{\theta,c_m}^{t_m} \circ \cdots \circ \psi_{\theta,c_1}^{t_1})(x)$. This allows us to determine the exact solution for $\phi^\theta(x,t)$. □

# D  Properties and Proofs

We present a summary of the properties offered by our DIGRAF that are absent in general-purpose activation functions or existing graph activations in Table 4.

Similar to Appendix C, for simplicity, in this Section, we drop the layer notations $l$.

In this section, we present the propositions and proofs for the properties outlined in Section 4.3. We begin by remarking that as shown in Section 4.3, DIGRAF is bounded within the domain $\Omega = [a, b]$, where $a < b$ by construction. We then present Proposition D.1 that outlines the Lipschitz constant of the velocity field $v^\theta$, followed by Proposition D.2, showing that DIGRAF is also Lipschitz continuous, and provide an upper bound on its Lipschitz constant.

**Proposition D.1** (The Lipschitz Constant of $v^{\boldsymbol{\theta}}$). *Given two arbitrary points $x, y \in \mathbb{R}$, and velocity field parameters $\boldsymbol{\theta} \in \mathbb{R}^{\mathcal{N}_\mathcal{P}-1}$ that define the continuous piecewise-affine velocity field $v^{\boldsymbol{\theta}}$, there exists a Lipschitz constant $C_{v^{\boldsymbol{\theta}}} = \sum_{j=0}^{\mathcal{N}_\mathcal{P}-2} |\boldsymbol{\theta}_j|$ such that*

$$\left| v^{\boldsymbol{\theta}}(x) - v^{\boldsymbol{\theta}}(y) \right| \leq C_{v^{\boldsymbol{\theta}}} \|(\tilde{\boldsymbol{x}} - \tilde{\boldsymbol{y}})\|_2, \tag{14}$$

*where $|\cdot|$ and $\|\cdot\|_2$ denote the absolute value of a scalar and the $\ell_2$ norm of a vector, respectively.*

*Proof.* First, we note that it was shown in Freifeld et al. [24, 25] that $v^{\boldsymbol{\theta}}$ is Lipschitz continuous, and now we provide a derivation of that Lipschitz constant. Following Definition C.2, the velocity field $v^{\boldsymbol{\theta}}$ is defined as $v^{\boldsymbol{\theta}}(x) = \sum_{j=0}^{\mathcal{N}_\mathcal{P}-2} \boldsymbol{\theta}_j \mathbf{b}_j \tilde{x}$, where $\{\mathbf{b}_j\}_{j=0}^{\mathcal{N}_\mathcal{P}-2}$ is an orthonormal basis of the velocity space. By the definition of $v^{\boldsymbol{\theta}}(x)$ and $v^{\boldsymbol{\theta}}(y)$, we have the following:

$$\left| v^{\boldsymbol{\theta}}(x) - v^{\boldsymbol{\theta}}(y) \right| = \left| \sum_{j=0}^{\mathcal{N}_\mathcal{P}-2} \boldsymbol{\theta}_j \mathbf{b}_j \tilde{\boldsymbol{x}} - \sum_{j=0}^{\mathcal{N}_\mathcal{P}-2} \boldsymbol{\theta}_j \mathbf{b}_j \tilde{\boldsymbol{y}} \right| \tag{15}$$

$$= \left| \sum_{j=0}^{\mathcal{N}_\mathcal{P}-0} \boldsymbol{\theta}_j \mathbf{b}_j (\tilde{\boldsymbol{x}} - \tilde{\boldsymbol{y}}) \right| \tag{16}$$

$$\leq \sum_{j=0}^{\mathcal{N}_\mathcal{P}-2} |\boldsymbol{\theta}_j| \|\mathbf{b}_j\|_2 \|(\tilde{\boldsymbol{x}} - \tilde{\boldsymbol{y}})\|_2 \tag{17}$$

$$= \sum_{j=0}^{\mathcal{N}_\mathcal{P}-2} |\boldsymbol{\theta}_j| \|(\tilde{\boldsymbol{x}} - \tilde{\boldsymbol{y}})\|_2 \tag{18}$$

$$= \|(\tilde{\boldsymbol{x}} - \tilde{\boldsymbol{y}})\|_2 \sum_{j=0}^{\mathcal{N}_\mathcal{P}-2} |\boldsymbol{\theta}_j| \tag{19}$$

$$= C_{v^{\boldsymbol{\theta}}} \|(\tilde{\boldsymbol{x}} - \tilde{\boldsymbol{y}})\|_2, \tag{20}$$

where the transition between Equation (16) and Equation (17) follows from the triangle inequality, and the transition between Equation (17) and Equation (18) follows from $\mathbf{b}_j$ being an orthonormal vector.

From the derivation above, and the fact that we know from Freifeld et al. [24, 25] that the velocity field is Lipschitz continuous, we conclude that the Lipschitz constant $C_{v^{\boldsymbol{\theta}}}$ of $v^{\boldsymbol{\theta}}$ reads $C_{v^{\boldsymbol{\theta}}} = \sum_{j=0}^{\mathcal{N}_\mathcal{P}-2} |\boldsymbol{\theta}_j|$. $\square$

Given the Lipschitz constant $C_{v^{\boldsymbol{\theta}}}$ for $v^{\boldsymbol{\theta}}$, we proceed to demonstrate that the transformation $T(\cdot; \boldsymbol{\theta})$ in DIGRAF is Lipschitz continuous, as well as bounding its Lipschitz constant.

**Proposition D.2** (The Lipschitz Constant of DIGRAF). *The diffeomorphic function $T(\cdot; \boldsymbol{\theta})$ in DIGRAF is defined in Equation (6) for a given set of weights $\boldsymbol{\theta}$, which in turn define the velocity field $v^{\boldsymbol{\theta}}$. Let $x, y \in \mathbb{R}$ be two arbitrary points, then the following inequality is satisfied:*

$$|T(x; \boldsymbol{\theta}) - T(y; \boldsymbol{\theta})| \leq |x - y| \exp(C_{v^{\boldsymbol{\theta}}}) \tag{21}$$

*where $C_{v^{\boldsymbol{\theta}}}$ is the Lipschitz constant of $v^{\boldsymbol{\theta}}$.*

*Proof.* We begin by substituting $T(\cdot; \boldsymbol{\theta})$ with Equation (1) and Equation (6). Utilizing Proposition D.1, we then establish an upper bound for $|T(x; \boldsymbol{\theta}) - T(y; \boldsymbol{\theta})|$ as follows:

$$|T(x; \boldsymbol{\theta}) - T(y; \boldsymbol{\theta})| = |x + \int_0^1 v^{\boldsymbol{\theta}}(\phi^{\boldsymbol{\theta}}(x, \tau)) \, d\tau - y - \int_0^1 v^{\boldsymbol{\theta}}(\phi^{\boldsymbol{\theta}}(y, \tau)) \, d\tau| \tag{22}$$

$$\leq |x - y| + C_{v^{\boldsymbol{\theta}}} \int_0^1 \left| (\phi^{\boldsymbol{\theta}}(x, \tau) - \phi^{\boldsymbol{\theta}}(y, \tau)) \right| \tag{23}$$

$$\leq |x - y| \exp(C_{v^{\boldsymbol{\theta}}}), \tag{24}$$

where $C_{v^{\boldsymbol{\theta}}}$ is the Lipschitz constant of $v^{\boldsymbol{\theta}}$ (Proposition D.1) and the last transition follows from Grönwall's inequality [30]. Consequently, the Lipschitz constant of DIGRAF is bounded from above by $\exp(C_{v^{\boldsymbol{\theta}}})$. $\square$

Now that we established that $T(\cdot; \boldsymbol{\theta})$ is Lipschitz continuous and presented an upper bound, we investigate what is the maximal difference in the output of $T(\cdot; \boldsymbol{\theta})$ with respect to two arbitrary inputs $x, y \in \Omega$, and whether it can be bounded. To address this, we present the following remark:

*Remark* D.3. Given a bounded domain $\Omega = [a, b]$, $a < b$, by construction, the diffeomorphism $T(\cdot; \boldsymbol{\theta})$ with parameter $\boldsymbol{\theta}$ in DiGRAF, as in Equation (7), is a $\Omega \to \Omega$ transformation [24, 25]. Therefore, by the max value theorem, the maximal output discrepancy for arbitrary $x, y \in \Omega$ is $|b - a|$, i.e., $|T(x; \boldsymbol{\theta}) - T(y; \boldsymbol{\theta})| \leq |b - a|$.

Combining the Proposition D.1, Proposition D.2 and Remark D.3, we formalize and prove the following proposition:

**Proposition 4.1** (The boundedness of $T(\cdot; \boldsymbol{\theta}^{(l)})$ in DiGRAF). *Given a bounded domain $\Omega = [a, b] \subset \mathbb{R}$ where $a < b$, and any two arbitrary points $x, y \in \Omega$, the maximal difference of a diffeomorphism $T(\cdot; \boldsymbol{\theta}^{(l)})$ with parameter $\boldsymbol{\theta}^{(l)}$ in DiGRAF is bounded as follows:*

$$|T(x; \boldsymbol{\theta}^{(l)}) - T(y; \boldsymbol{\theta}^{(l)})| \leq \min(|b - a|, |x - y| \exp(C_{v^{\boldsymbol{\theta}^{(l)}}})) \tag{11}$$

*where $C_{v^{\boldsymbol{\theta}^{(l)}}}$ is the Lipschitz constant of the CPA velocity field $v^{\boldsymbol{\theta}^{(l)}}$.*

*Proof.* In Proposition D.2 we presented an upper bound on the Lipschitz constant of $T(\cdot; \boldsymbol{\theta})$, and in D.3 we also presented an upper bound on the maximal difference between the application of $T(\cdot; \boldsymbol{\theta})$ on two inputs $x, y$. Combining the two bounds, we get the following inequality:

$$|T(x; \boldsymbol{\theta}) - T(y; \boldsymbol{\theta})| \leq \min(|b - a|, |x - y| \exp(C_{v^{\boldsymbol{\theta}}})). \tag{25}$$

$\square$

The result in Proposition 4.1 gives us a tighter upper bound on the boundedness of the transformation $T(\cdot; \boldsymbol{\theta})$ in our DiGRAF that is related both to the hyperparameters $a, b$, as well as the learned velocity field parameters $\boldsymbol{\theta}$.

Next, we discuss another property outlined in Section 4.3, demonstrating that DiGRAF is permutation equivariant – a desirable property when designing a GNN component [8].

**Proposition D.4** ( DiGRAF is permutation equivariant.). *Consider a graph encoded by the adjacency matrix $\mathbf{A} \in \mathbb{R}^{N \times N}$, where $N$ is the number of nodes. Let $\bar{\mathbf{H}}^{(l)} \in \mathbb{R}^{N \times C}$ be the intermediate node features at layer $l$, before the element-wise application of our DiGRAF. Let $\mathbf{P}$ be an $N \times N$ permutation matrix. Then,*

$$\text{DiGRAF}(\mathbf{P}\bar{\mathbf{H}}^{(l)}, \boldsymbol{\theta}_P^{(l)}) = \mathbf{P}\,\text{DiGRAF}(\bar{\mathbf{H}}^{(l)}, \boldsymbol{\theta}^{(l)}) \tag{26}$$

*where $\boldsymbol{\theta}_P^{(l)}$ and $\boldsymbol{\theta}^{(l)}$ are obtained by feeding $\mathbf{P}\bar{\mathbf{H}}^{(l)}$ and $\bar{\mathbf{H}}^{(l)}$, respectively, to Equation (9).*

*Proof.* We break down the proof into two parts. First, we show that $\text{GNN}_{\text{ACT}}$ outputs the same $\boldsymbol{\theta}$ under permutations, that is we show

$$\boldsymbol{\theta}_P^{(l)} = \boldsymbol{\theta}^{(l)}.$$

Second, we prove that the activation function $T^{(l)}$ is permutation equivariant, ensuring the overall method maintains this property.

To begin with, recall that Equation (9) is composed by $\text{GNN}_{\text{ACT}}$, which is permutation equivariant, and by a pooling layer, which is permutation invariant. Therefore their composition is permutation invariant, that is $\boldsymbol{\theta}_P^{(l)} = \boldsymbol{\theta}^{(l)}$.

Prior to the activation function layer $T^{(l)}$, $\bar{\mathbf{H}}^{(l)}$ undergoes rescaling as described in Appendix B, which is permutation equivariant as it operates element-wise. Finally, since activation function $T^{(l)}$ acts element-wise, and given that $\boldsymbol{\theta}$ remains unchanged, the related CPA velocity fields are identical, resulting in the same transformed output for each entry, despite the entries being permuted in $\mathbf{P}\bar{\mathbf{H}}^{(l)}$. Therefore, DiGRAF is permutation equivariant. $\square$

## D.1 Diffeomorphic Activation Functions

In this section, we provide several examples of popular and well-known diffeomorphic functions, contributing to our motivation to utilize diffeomorphisms as a blueprint for learning graph activation functions. We remark that, differently from standard activation functions, our DiGRAF does not need to follow a predefined, fixed template, but can instead learn a diffeomorphism best suited for the task and input, as $T^{(l)}$ within CPAB can represent a wide range of diffeomorphisms [24, 25].

We recall that, as outlined in Section 3.1, a function is classified as a diffeomorphism if it is (1) bijective, (2) differentiable, and (3) has a differentiable inverse.

**Sigmoid.** We denote the Sigmoid activation function as $\sigma : \mathbb{R} \to (0, 1)$, defined by

$$\sigma(x) = \frac{1}{1 + e^{-x}}.$$

To prove that $\sigma$ is a diffeomorphism, we first establish its bijectivity. Injectivity follows from observing that for any distinct points $x_1$ and $x_2$ in $\mathbb{R}$, $\sigma(x_1) = \frac{1}{1+e^{-x_1}}$ can only equal $\sigma(x_2) = \frac{1}{1+e^{-x_2}}$ if and only if $x_1 = x_2$. For surjectivity, we represent $x$ as a function of $y$, such that $y = \frac{1}{1+e^{-x}} \implies x = -\ln\left(\frac{1-y}{y}\right)$, ensuring that for every $y \in (0,1)$ there is an element $x \in \mathbb{R}$ such that $\sigma(x) = y$.

To demonstrate differentiability, we examine the derivative of $\sigma$. The derivative

$$\frac{d}{dx}\sigma(x) = \sigma(x)(1 - \sigma(x)),$$

which is continuous. Additionally, the inverse function

$$\sigma^{-1}(y) = -\ln\left(\frac{1-y}{y}\right)$$

is also bijective and differentiable. Thus, with all these requirements satisfied, $\sigma$ is indeed a diffeomorphism.

**Tanh.** The hyperbolic tangent function

$$\tanh(x) = \frac{e^x - e^{-x}}{e^x + e^{-x}}$$

is a diffeomorphism from $\mathbb{R}$ to $(-1, 1)$. To establish this, we demonstrate that $\tanh$ is bijective and differentiable, with a differentiable inverse function.

Firstly, $\tanh$ is injective because if $\tanh(x_1) = \tanh(x_2)$, then $x_1 = x_2$. It is also surjective because for any $y \in (-1, 1)$, there exists $x = \frac{1}{2}\ln\left(\frac{1+y}{1-y}\right)$ such that $\tanh(x) = y$.

The derivative

$$\frac{d}{dx}\tanh(x) = 1 - \tanh^2(x)$$

is continuous and positive. Additionally, the inverse function

$$\tanh^{-1}(y) = \frac{1}{2}\ln\left(\frac{1+y}{1-y}\right)$$

is continuously differentiable. Therefore, $\tanh$ qualifies as a diffeomorphism.

**Softplus.** To establish the Softplus function

$$\text{softplus}(x) = \ln(1 + e^x)$$

as a diffeomorphism from $\mathbb{R}$ to $(0, \infty)$, we first demonstrate its injectivity and surjectivity.

Assuming $\text{softplus}(x_1) = \text{softplus}(x_2)$, we obtain $e^{x_1} = e^{x_2}$, implying $x_1 = x_2$, hence establishing injectivity. For any $y \in (0, \infty)$, we find an $x \in \mathbb{R}$ such that $y = \ln(1 + e^x)$, ensuring surjectivity.

The derivative of the Softplus function,

$$\frac{d}{dx}\text{softplus}(x) = \frac{e^x}{1 + e^x} = \sigma(x),$$

where $\sigma(x)$ is the Sigmoid function, known to be continuous and differentiable. Therefore, $\text{softplus}(x)$ is continuously differentiable.

Considering the inverse of the Softplus function,

$$\text{softplus}^{-1}(y) = \ln(e^y - 1),$$

its derivative is

$$\frac{d}{dy}\text{softplus}^{-1}(y) = \frac{e^y}{e^y - 1},$$

which is continuous for all $y > 0$, indicating that $\text{softplus}^{-1}(y)$ is continuously differentiable for all $y > 0$. Therefore, we conclude that the softplus function qualifies as a diffeomorphism.

**ELU.** The ELU activation function [11] is defined as below:

$$\text{ELU}(x) = \begin{cases} x & \text{if } x > 0 \\ \alpha(e^x - 1) & \text{if } x \leq 0 \end{cases}$$

where $\alpha \in \mathbb{R}$ is a constant that scales the negative part of the function. To demonstrate that ELU is bijective, we analyze its injectivity and surjectivity. For $x > 0$, ELU acts as the identity function, which is inherently injective. For $x \leq 0$, $\alpha(e^{x_1} - 1) = \alpha(e^{x_2} - 1)$, implies $x_1 = x_2$. The inverse function for ELU is given by:

$$\text{ELU}^{-1}(y) = \begin{cases} y & \text{if } y > 0 \\ \ln(\frac{y}{\alpha} + 1) & \text{if } y \leq 0 \end{cases}$$

This inverse maps every value in the codomain back to a unique value in the domain, proving that ELU is surjective.

Next, we examine the continuity of ELU. At $x = 0$, $\text{ELU}(x = 0) = \alpha(e^0 - 1) = 0$. Next, we check the limits for both sides of 0. For $x > 0$, $\lim_{x \to 0^+} \text{ELU}(x) = \lim_{x \to 0^+} x = 0$, while for $x \leq 0$, we have $\lim_{x \to 0^-} \text{ELU}(x) = \lim_{x \to 0^-} \alpha(e^x - 1) = 0$. Since both limits are equal, the ELU function is continuous at $x = 0$. For the derivative of ELU, i.e.,

$$\frac{d}{dx}\text{ELU}(x) = \begin{cases} 1 & \text{if } x > 0 \\ \alpha e^x & \text{if } x \leq 0 \end{cases}$$

at $x = 0$, we have $\frac{d}{dx}\text{ELU}(x) = \alpha e^0 = \alpha$. By setting $\alpha = 1$, the derivative at $x = 0$ matches the derivative for $x > 0$, making the derivative continuous.

The derivative for the inverse function is

$$\frac{d}{dy}\text{ELU}^{-1}(y) = \begin{cases} 1 & \text{if } y > 0 \\ \frac{1}{y+\alpha} & \text{if } y \leq 0 \end{cases}$$

which is also continuously differentiable. Hence, ELU is a diffeomorphism.

# E   Additional Results

## E.1   Function Approximation with CPAB

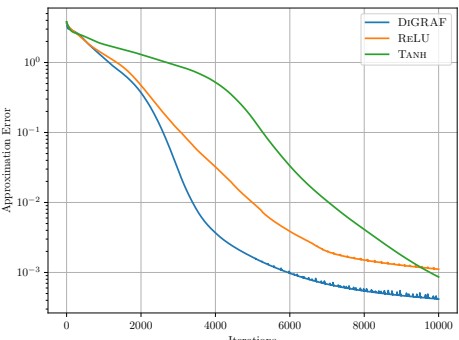

Figure 6: The approximation error of the Peaks function (Equation (27)) with ReLU, Tanh, and DIGRAF.

The combination of learned linear layers together with non-linear functions such as ReLU and Tanh are well-known to yield good function approximations [13, 14]. Therefore, when designing an activation function blueprint, i.e., the template by which the activation function is learned, it is important to consider its approximation power. In Section 1 and in particular in Figure 2, we demonstrate the ability of the CPAB framework to approximate known activation functions. We now show additional evidence for the flexibility and power of CPAB as a framework for learning activation functions, leading to our DIGRAF. To this end, we consider the ability of a multilayer perceptron (MLP) with various activation functions (ReLU, Tanh, and DIGRAF) to approximate the well-known *'peaks'* function that mathematically reads:

$$g(x,y) = 3(1-x)^2\exp(-(x^2)-(y+1)^2) - 10(\frac{x}{5}-x^3-y^5)\exp(-x^2-y^2) - \frac{1}{3}\exp(-(x+1)^2-y^2). \quad (27)$$

The peaks function in Equation (27) is often times used to measure the ability of methods to approximate functions [32], where the input is point pairs $(x, y) \in \mathbb{R}^2$, and the goal is to minimize the mean-squared-error between the predicted function value $g$ and the actual function value $x$. Formally, we consider the following MLP:

$$\hat{g}(x,y) = (\sigma(\sigma(\begin{bmatrix} x, y \end{bmatrix} W_1)W_2)W_3), \quad (28)$$

where $\sigma$ is the activation of choice (ReLU, Tanh, or DIGRAF), and $W_1 \in \mathbb{R}^{2 \times 64}$, $W_2 \in \mathbb{R}^{64 \times 64}$, $W_3 \in \mathbb{R}^{64 \times 1}$ are the trainable parameter matrices of the linear layers in the MLP. The goal, as discussed above, is to minimize the loss $\|\hat{g}(x,y) - g(x,y)\|_2$, for data triplets $(x_i, y_i, g(x_i, y_i))$ sampled from the peaks function. In our experiment, we sample 50,000 points, and report the obtained approximation error in terms of MSE in Figure 6. As can be seen, our DIGRAF, based on the CPAB framework, allows to obtain a significantly lower approximation error, up to 10 times lower (better) than ReLU, and 3 times better than Tanh. This example further motivates us to harness CPAB as the blueprint of DIGRAF.

Table 5: Graph classification accuracy (%) ↑ on TUDatasets. The top three methods are marked by **First**, **Second**, **Third**.

| Method ↓ / Dataset → | MUTAG | PTC | PROTEINS | NCI1 | NCI109 |
|---|---|---|---|---|---|
| **STANDARD ACTIVATIONS** | | | | | |
| GIN + Identity | 91.4±5.6 | 66.2±5.5 | 75.9±3.2 | 82.8±2.0 | 82.8±1.3 |
| GIN + Sigmoid [69] | 90.9±5.5 | 65.3±4.8 | 75.0±5.0 | 82.6±1.4 | 81.2±1.6 |
| GIN + ReLU [84] | 89.4±5.6 | 64.6±7.0 | 76.2±2.8 | 82.7±1.7 | 82.2±1.6 |
| GIN + LeakyReLU [50] | 90.9±5.7 | 65.0±9.0 | 76.2±4.4 | **83.5±2.2** | 82.9±2.0 |
| GIN + Tanh [35] | **92.5±7.9** | 65.1±6.5 | 75.9±4.3 | 83.2±2.6 | **83.0±2.6** |
| GIN + GeLU [34] | 90.9±7.0 | 65.4±7.9 | 76.6±2.8 | **83.5±1.4** | 82.9±1.6 |
| GIN + ELU [12] | **92.5±5.6** | 65.4±7.5 | 75.4±2.7 | 83.3±2.0 | 82.6±1.7 |
| **LEARNABLE ACTIVATIONS** | | | | | |
| GIN + PReLU [33] | 91.7±6.7 | **66.9±7.0** | **76.7±3.5** | 82.9±2.6 | 82.3±1.8 |
| GIN + Maxout [29] | 91.5±7.5 | 66.8±8.3 | **76.8±4.0** | 83.3±2.9 | **83.0±3.0** |
| GIN + Swish [66] | 90.4±4.8 | 65.1±6.3 | 76.2±4.2 | **83.4±1.4** | 82.9±3.0 |
| **GRAPH ACTIVATIONS** | | | | | |
| GIN + Max [38] | 90.9±7.1 | **67.7±9.2** | 75.9±3.1 | 83.3±2.0 | 82.7±1.9 |
| GIN + Median [38] | **92.0±6.6** | **67.7±4.5** | 75.0±4.3 | **83.6±1.9** | 82.8±1.8 |
| GIN + GReLU [92] | **92.0±7.3** | 64.9±6.6 | **76.8±3.5** | 82.8±2.5 | 82.4±2.2 |
| GIN + DIGRAF (W/O ADAP.) | 92.0±5.6 | **68.9±7.5** | 77.2±3.6 | 83.0±1.3 | 82.9±2.2 |
| GIN + DIGRAF | **92.1±7.9** | 68.6±7.4 | **77.9±3.4** | **83.4±1.2** | **83.3±1.9** |

Table 6: Different GNN architectures (GCN, GAT, GIN, SAGE) coupled with ReLU, DIGRAF (W/O ADAP.) and DIGRAF activation functions. The top performing model is marked with the corresponding color for each architecture.

| Activation | Model | CORA ↑ | CITESEER ↑ | PUBMED ↑ | BLOG CATALOG ↑ | FLICKR ↑ | ZINC ↓ | MOLHIV ↑ |
|---|---|---|---|---|---|---|---|---|
| ReLU | GCN | 79.2±1.4 | 67.7±2.3 | 77.6±2.2 | 72.1±1.9 | 50.7±2.3 | 0.3674±0.0111 | 76.06±0.97 |
| | GAT | 78.0±2.1 | 63.6±1.9 | 77.0±1.7 | 74.2±1.8 | 55.5±1.1 | 0.3842±0.0070 | 76.00±0.82 |
| | GIN | 67.1±3.0 | 58.8±2.2 | 68.4±2.7 | 72.6±2.5 | 43.1±2.6 | 0.1630±0.0040 | 75.58±1.40 |
| | SAGE | 78.5±1.6 | 67.4±1.8 | 76.2±1.8 | 84.9±3.1 | 43.5±2.8 | 0.4680±0.0030 | 77.46±0.91 |
| DIGRAF (W/O ADAP.) | GCN | 81.5±1.1 | 69.2±2.1 | 78.3±1.6 | 80.8±0.6 | 68.6±1.8 | 0.3187±0.0083 | 76.62±1.20 |
| | GAT | 81.0±2.1 | 69.3±1.7 | 78.2±2.0 | 79.3±2.8 | 62.8±6.9 | 0.3309±0.0115 | 76.80±1.14 |
| | GIN | 80.6±2.3 | 67.5±4.2 | 76.0±4.0 | 82.1±3.5 | 68.0±1.3 | 0.1382±0.0082 | 79.19±1.36 |
| | SAGE | 79.3±7.8 | 67.7±2.5 | 77.1±3.2 | 90.6±0.3 | 66.5±6.5 | 0.4442±0.0097 | 78.19±0.83 |
| DIGRAF | GCN | 82.8±1.1 | 69.5±1.4 | 79.3±1.4 | 81.6±0.8 | 69.6±0.6 | 0.2830±0.0054 | 77.38±2.31 |
| | GAT | 81.0±1.5 | 69.4±2.4 | 78.9±2.5 | 79.3±4.2 | 62.9±1.0 | 0.2918±0.0133 | 77.47±1.18 |
| | GIN | 80.6±2.0 | 68.9±3.8 | 76.9±3.3 | 83.0±4.1 | 70.6±4.3 | 0.1302±0.0090 | 80.28±1.44 |
| | SAGE | 79.9±6.9 | 68.0±4.0 | 77.3±3.3 | 90.8±0.4 | 69.0±4.8 | 0.4147±0.0078 | 79.32±0.74 |

## E.2 Results on TUDatasets

Our results are summarized in Table 5, where we consider MUTAG, PTC, PROTEINS, NCI1 and NCI109 datasets from the TU repository [56]. As can be seen from the table, DIGRAF is consistently among the top-3 best-performing activation functions, and it consistently outperforms other graph-adaptive activation functions. These results support our design choices for DIGRAF and the flexibility offered by CPAB diffeomorphisms.

## E.3 Comparison with Different GNN Architectures

We now provide a comparison between DIGRAF and the ReLU activation function coupled with GCN, GAT, GIN, and SAGE backbones in Table 6. Notably, DIGRAF consistently outperforms ReLU regardless of the backbone architecture.

## E.4 Visualization of DIGRAF

To gain a qualitative understanding of the behavior of DIGRAF, we now illustrate the activation function learned by DIGRAF after the last GNN layer on different graphs. To this end, we randomly selected two graphs from the ZINC dataset, as shown in Figure 7. The original graphs are presented in the lower right, with each color representing a feature. Nodes with the same color share the same feature. The comparison of the figures demonstrates that for different graphs, with different features and structures, DIGRAF learns distinct activation functions, showing its adaptivity to the input graph.

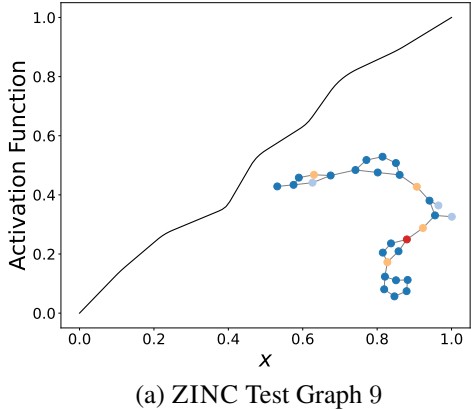
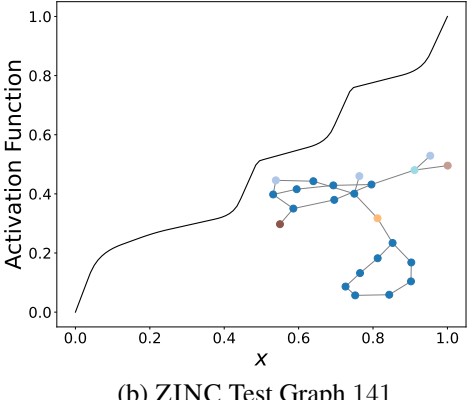

(a) ZINC Test Graph 9            (b) ZINC Test Graph 141

Figure 7: Activation function learned by DIGRAF after the last GNN layer on two randomly selected graphs from ZINC. Different node colors indicate different node features. DIGRAF yields different activations for different graphs.

Table 7: Performance Comparison of DIGRAF with ReLU variants of increased parameter budget. The number of parameters is reported within the parenthesis adjacent to the metric. We use GINE [37] as a backbone. Increasing the parameter count with ReLU does not yield significant improvements, and DIGRAF outperforms all variants, even those with a higher number of parameters. Note that, DIGRAF (W/O ADAP.) has only $\mathcal{N}_\mathcal{P} - 1$ additional parameters, where $\mathcal{N}_\mathcal{P}$ is the tessellation size.

| Method ↓ / Dataset → | ZINC (MAE ↓) | MOLHIV (ACC. % ↑) |
|---|---|---|
| GIN + ReLU (standard) | $0.1630 \pm 0.0040$ ($\sim$ 308K) | $75.58 \pm 1.40$ ($\sim$ 63K) |
| GIN + ReLU (double #channels) | $0.1578 \pm 0.0014$ ($\sim$ 1207K) | $75.73 \pm 0.71$ ($\sim$ 240K) |
| GIN + ReLU (double #layers) | $0.1609 \pm 0.0033$ ($\sim$ 580K) | $75.78 \pm 0.43$ ($\sim$ 116K) |
| DIGRAF (W/O ADAP.) | $0.1382 \pm 0.0080$ ($\sim$ 308K) | $79.19 \pm 1.36$ ($\sim$ 63K) |
| DIGRAF | $\mathbf{0.1302 \pm 0.0090}$ ($\sim$ 333K) | $\mathbf{80.28 \pm 1.44}$ ($\sim$ 83K) |

## E.5 Parameter Count Comparison

GNN$_{\text{ACT}}$ is a core component of DIGRAF, which ensures graph-adaptivity by generating the parameters $\boldsymbol{\theta}^{(l)}$ of the activation function conditioned on the input graph. While the benefits of graph-adaptive activation functions are evident from our experiments in Section 5, as DIGRAF consistently outperforms DIGRAF (W/O ADAP.), the variant of our method that is not graph adaptive, it comes at the cost of additional parameters to learn GNN$_{\text{ACT}}$ (Equation (9)). Specifically, because in all our experiments GNN$_{\text{ACT}}$ is composed of 2 layers and a hidden dimension of 64, DIGRAF adds at most approximately 20K additional parameters. The number of added parameters in DIGRAF (W/O ADAP.) is significantly lower, counting at $\mathcal{N}_\mathcal{P} - 1$, where $\mathcal{N}_\mathcal{P}$ is the tessellation size. Note in our experiments, the tessellation size does not exceed 16. To further understand whether the improved performance of DIGRAF is due to the increased number of parameters, we conduct an additional experiment using the ReLU activation function where we increase the number of parameters of the model and compare the performances. In particular, we consider following settings: (1) The standard variant (GIN + ReLU), (2) The variant obtained by doubling the number of layers, and (3) The variant is obtained by doubling the number of hidden channels.

We present the results of the experiment described above on the ZINC-12K and MOLHIV datasets in Table 7. We observed that adding more parameters to the ReLU baseline does not produce significant performance improvements, even in cases where the baselines have $\sim$4 times more parameters than DIGRAFand its baseline. On the contrary, with DIGRAF significantly improved performance is obtained compared to the baselines.

## F Ablation Studies

We present the impact of several key components of DIGRAF, namely the tessellation size $\mathcal{N}_\mathcal{P}$, the depth of GNN$_{\text{ACT}}$ (Equation (9)) and the regularization coefficient $\lambda$ of $\boldsymbol{\theta}^{(l)}$ (Equation (12)). We choose a few

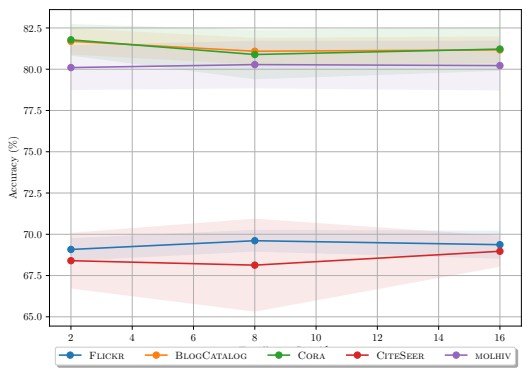

Figure 8: Impact of tessellation size $\mathcal{N}_\mathcal{P}$ on the performance of DIGRAF on CORA, CITESEER, FLICKR, BLOGCATALOG, and MOLHIV datasets.

Table 8: Effect of depth of $\text{GNN}_{\text{ACT}}$ on DIGRAF.

| Dataset | $L_{\text{ACT}} = 2$ | $L_{\text{ACT}} = 4$ | $L_{\text{ACT}} = 6$ |
|---|---|---|---|
| FLICKR | 69.6±0.6 | 66.3±0.8 | 69.3±0.7 |
| BLOGCATALOG | 81.0±0.5 | 81.1±0.48 | 81.6±0.8 |
| MOLHIV | 80.28±1.44 | 80.19±1.49 | 80.22±1.56 |
| ZINC | 0.1302±0.0090 | 0.1309±0.0084 | 0.1314±0.083 |

representative datasets, i.e., CORA, CITESEER, FLICKR and BLOGCATALOG for which we use GCN [43]; and ZINC-12K and MOLHIV for which we use GINE [37] as GNN respectively.

## F.1  Tessellation Size

Recall that the tessellation size $\mathcal{N}_\mathcal{P}$ determines the dimension of $\boldsymbol{\theta}^{(l)} \in \mathbb{R}^{\mathcal{N}_\mathcal{P}-1}$ that parameterizes the velocity fields within DIGRAF. We study the effect of the tessellation size on the performance of DIGRAF in Figure 8. We can see that a small tessellation size is sufficient for good performance, and increasing its size results in marginal changes. This observation suggests that CPAB is highly flexible, and aligns with the conclusions in previous studies on different applications of CPAB [52], which have shown that small sizes are sufficient in most cases.

## F.2  Depth of $\text{GNN}_{\text{ACT}}$

DIGRAF exhibits graph adaptivity by predicting $\boldsymbol{\theta}^{(l)} \in \mathbb{R}^{\mathcal{N}_\mathcal{P}-1}$ conditioned on the input graph through $\text{GNN}_{\text{ACT}}$. Table 8 shows the impact of the number of layers $L_{\text{ACT}}$ of $\text{GNN}_{\text{ACT}}$ on the performance of DIGRAF. In particular, we maintain a fixed architecture for DIGRAF and vary only $L_{\text{ACT}}$. The results show that increasing the depth of $\text{GNN}_{\text{ACT}}$ improves only marginally the performance of DIGRAF, demonstrating that the increased number of parameters is not the main factor of the better performance of DIGRAF. On the contrary, the flexibility and adaptivity offered by DIGRAF are the main factors of the improvements, as demonstrated by DIGRAF consistently outperforming DIGRAF (W/O ADAP.) and other activation functions (Section 5).

## F.3  Regularization

As discussed in Section 4.2, the regularization enforces the smoothness of the velocity field. We investigate the impact of the value of the regularization coefficient $\lambda$ on DIGRAF (Equation (12)) in Table 9. The results reveal that the optimal value of $\lambda$ depends on the dataset of interest, with small positive values yielding generally good results across all datasets.

## F.4  Comparison of DIGRAF and DIGRAF (W/O ADAP.) with Equal Parameter Budget

To demonstrate the efficacy of graph adaptivity provided by $\text{GNN}_{\text{ACT}}$, we conduct an experiment where we increase the number of layers and channels of $\text{GNN}_{\text{LAYER}}$ in DIGRAF (W/O ADAP.) to match the total number of parameters in DIGRAF. As shown in Table 10, the increase in the number of parameters does not translate to better performance. Rather, the effective usage of the extra parameters as done by $\text{GNN}_{\text{ACT}}$ is the reason behind the performance boost offered by DIGRAF.

Table 9: Effect of velocity field regularization coefficient $\lambda$ on DiGRAF.

| Dataset | $\lambda = 0.0$ | $\lambda = 0.001$ | $\lambda = 0.01$ | $\lambda = 1.0$ |
|---|---|---|---|---|
| Flickr (% Acc ↑) | 69.1±0.9 | 68.7±0.7 | 69.6±0.6 | 69.0±0.9 |
| BlogCatalog (% Acc ↑) | 80.5±0.9 | 81.0±0.8 | 81.4±1.0 | 81.6±0.8 |
| molhiv (% Acc ↑) | 79.38±2.10 | 80.28±1.44 | 80.16±1.50 | 78.15±1.29 |
| ZINC (MAE ↓) | 0.1395±0.0102 | 0.1348±0.0093 | 0.1302±0.0090 | 0.1353±0.0071 |

Table 10: Results on ZINC and molhiv datasets along with number of parameters in parenthesis.

| Method | ZINC (MAE) ↓ | MOLHIV (ROC AUC) ↑ |
|---|---|---|
| GIN + DiGRAF (W/O Adap.) with larger GNN$_{\text{Layer}}$ | 0.1388 ± 0.0071 (337K) | 79.22 ± 1.40 (85K) |
| GIN + DiGRAF (W/O Adap.) (Original) | 0.1382 ± 0.0086 (308K) | 79.19 ± 1.36 (63K) |
| GIN + DiGRAF | 0.1302 ± 0.0094 (333K) | 80.28 ± 1.44 (83K) |

# G  Experimental Details

We implemented DiGRAF using PyTorch [63] (offered under BSD-3 Clause license) and the PyTorch Geometric library [22] (offered under MIT license). All experiments were conducted on NVIDIA RTX A5000, NVIDIA GeForce RTX 4090, NVIDIA GeForce RTX 4070 Ti Super, NVIDIA GeForce GTX 1080 Ti, NVIDIA TITAN RTX and NVIDIA TITAN V GPUs. For hyperparameter tuning and model selection, we utilized the Weights and Biases (wandb) library [6]. We used the difw package [52, 25, 24, 74] (offered under MIT license) for the diffeomorphic transformations based on the closed-form integration of CPA velocity functions. In the following subsections, we present the experimental procedure, dataset details, and hyperparameter configurations for each task.

**Hyperparameters.** The hyperparameters include the number of layers $L$ and embedding dimension $C$ of $\text{GNN}_{\text{LAYER}}^{(l)}$, learning rates and weight decay factors for both $\text{GNN}_{\text{LAYER}}^{(l)}$ and $\text{GNN}_{\text{ACT}}$, dropout rate $p$, tessellation size $\mathcal{N}_{\mathcal{P}}$, and regularization coefficient $\lambda$. We additionally include the number of layers $L_{\text{ACT}}$ and embedding dimension $C_{\text{ACT}}$ of $\text{GNN}_{\text{ACT}}$. We employed a combination of grid search and Bayesian optimization. All hyperparameters were chosen according to the best validation metric. For the baselines, we include only the applicable hyperparameters in our search space.

**Node Classification.** For each dataset, we train a 2-layer GCN [43] as the backbone architecture, and integrate each of the activation functions into this model. Following Zhang et al. [92], we randomly choose 20 nodes from each class for training and select 1000 nodes for testing. For each activation function, we run the experiment 10 times with random partitions. We report the mean and standard deviation of node classification accuracy on the test set. Table 11 summarizes the statistics of the node classification datasets used in our experiments. All models were trained for 1000 epochs with a fixed batch size of 32 using the Adam optimizer. Tables 12 and 13 lists the hyperparameters and their search ranges or values.

**Graph Classification.** The statistics of various datasets can be found in Table 14. We consider the following setup:

- **ZINC-12k:** We consider the splits provided in Dwivedi et al. [17]. We use the mean absolute error (MAE) both as the loss and evaluation metric and report the mean and standard deviation over the test set calculated using five different seeds. We use the Adam optimizer and decay the learning rate by 0.5 every 300 epochs, with a maximum of 1000 epochs. In all our experiments, we adhere to the 500k parameter budget [17]. We use GINE [37] layers both for $\text{GNN}_{\text{LAYER}}^{(l)}$ and within $\text{GNN}_{\text{ACT}}$, and we fix $C_{\text{ACT}} = 64$ and $L_{\text{ACT}} = 2$. We report the hyperparameter search space for all the other hyperparameters in Table 15.

- **TUDatasets**: We follow the standard procedure prescribed in Xu et al. [84] for evaluation. That is, we use a 10 fold cross-validation and report the mean and standard deviation of the accuracy at the epoch that yields the best validation performance on average. We use the Adam optimizer and train for a maximum of 350 epochs. We use GIN [84] layers both for $\text{GNN}_{\text{LAYER}}^{(l)}$ and within $\text{GNN}_{\text{ACT}}$, and we fix $L_{\text{ACT}} = 2$. We present the hyperparameter search space for all other parameters in Table 15.

- **OGB:** We consider 4 datasets from the OGB repository, with one, namely molesol, being a regression problem, while the others are classification tasks. We run each experiment using five different seeds and report the mean and standard deviation of RMSE/ROC-AUC. We use the Adam optimizer, decaying the learning rate by a factor of 0.5 every 100 epochs, and train for a maximum of 500 epochs. We use the GINE model with the encoders prescribed in Hu et al. [36] both for $\text{GNN}_{\text{LAYER}}^{(l)}$

Table 11: Statistics of the node classification datasets [53, 73, 58, 86].

| Dataset | #nodes | #edges | #features | #classes |
|---|---|---|---|---|
| **PLANETOID** | | | | |
| CORA | 2,708 | 10,556 | 1,433 | 7 |
| CITESEER | 3,327 | 9,104 | 3,703 | 6 |
| PUBMED | 19,717 | 88,648 | 500 | 3 |
| **SOCIAL NETWORKS** | | | | |
| FLICKR | 7,575 | 479,476 | 12,047 | 9 |
| BLOGCATALOG | 5,196 | 343,486 | 8,189 | 6 |

Table 12: Hyperparameter configurations for the Planetoid datasets [53, 73, 58].

| Hyperparameter | Search Range / Value |
|---|---|
| Learning rate for $\text{GNN}_{\text{LAYER}}^{(l)}$ | $[10^{-5}, 10^{-4}, 10^{-3}, 5 \times 10^{-3}, 5 \times 10^{-2}]$ |
| Learning rate for $\theta^{(l)}$ / $\text{GNN}_{\text{ACT}}$ | $[10^{-6}, 5 \times 10^{-6}, 10^{-5}, 10^{-4}, 10^{-3}, 5 \times 10^{-3}]$ |
| Weight decay | $[10^{-5}, 10^{-4}, 5 \times 10^{-3}, 0.0]$ |
| $C$ | $[64, 128, 256]$ |
| $C_{\text{ACT}}$ | $[64, 128]$ |
| $L_{\text{ACT}}$ | $[2, 4]$ |
| $p$ | $[0.0, 0.5]$ |
| $\mathcal{N}_{\mathcal{P}}$ | $[2, 4, 8, 16]$ |
| $\lambda$ | $[0.0, 10^{-3}, 10^{-2}, 1.0]$ |

and within $\text{GNN}_{\text{ACT}}$, and we set $C_{\text{ACT}} = 64$ and $L_{\text{ACT}} = 2$. We present the hyperparameter search space for all other parameters in Table 15

# H Complexity and Runtimes

**Time Complexity.** We now provide an analysis of the time complexity of DIGRAF. Let us recall the following details: (i) As described in Equation (8), DIGRAF is applied element-wise in parallel for each dimension of the output of $\text{GNN}_{\text{LAYER}}^{(l)}$. (ii) As described in Equation (9), we employ an additional GNN denoted by $\text{GNN}_{\text{ACT}}$ to compute $\theta^{(l)}$. In all our experiments, both the backbone GNN and $\text{GNN}_{\text{ACT}}$ are message-passing neural networks (MPNNs) [28]. (iii) As described in Theorem 2 of Freifeld et al. [25], for 1-dimensional domain, there exists a closed form for $T^{(l)}(\cdot; \theta^{(l)})$, and the complexity for the CPAB computations are linear with respect to the tesselation size, which is a constant of up to 16 in our experiments. Therefore, using DIGRAF with any linear complexity (with respect to the number of nodes and edges) MPNN-based backbone maintains the linear complexity of the backbone MPNN. Put precisely, each MPNN layer has linear complexity in the number of nodes $|V|$ and $|E|$. We use $L_{\text{ACT}}$ layers in $\text{GNN}_{\text{ACT}}$, the computational complexity of a DIGRAF layer is $\mathcal{O}(L_{\text{ACT}} \cdot (|V| + |E|))$. Since we have $L$ layers in overall GNN, the computational complexity of an MPNN-based GNN coupled with DIGRAF is $\mathcal{O}(L \cdot L_{\text{ACT}} \cdot (|V| + |E|))$. In our experiments, we fix the hyperparameter $L_{\text{ACT}} = 2$, resulting in $\mathcal{O}(L \cdot (|V| + |E|))$ computational complexity in practice.

**Memory Complexity.** DIGRAF uses $\text{GNN}_{\text{ACT}}$ which is an MPNN and hence has linear space complexity (with respect to the number of nodes and edges). CPAB computations require constant memory with respect to the graph size for a 1-dimensional domain due to the analytical implementation. We use $L$ layers in overall GNN and $L_{\text{ACT}}$ layers in $\text{GNN}_{\text{ACT}}$ resulting in a memory complexity of $\mathcal{O}(L \cdot L_{\text{ACT}} \cdot (|V| + |E|))$. In our experiments, we fix the hyperparameter $L_{\text{ACT}} = 2$, resulting in $\mathcal{O}(L \cdot (|V| + |E|))$ memory complexity in practice.

**Runtimes.** Despite having linear computational complexity in the size of the graph, DIGRAF performs additional computations to obtain $\theta^{(l)}$ using $\text{GNN}_{\text{ACT}}$. To understand the impact of these computations, we measured the training and inference times of DIGRAF and present it in Table 16. Specifically, we report the average time per batch and standard deviation of the same measured on an NVIDIA A5000 GPU, using a batch size of 128. For a fair comparison, we use the same number of layers, batch size, and channels in all methods. Additionally, for our DIGRAF, we set the number of layers within $\text{GNN}_{\text{ACT}}$ to $L_{\text{ACT}} = 2$, and the embedding dimension to $C_{\text{ACT}} = 64$. Our analysis indicates that while DIGRAF requires additional computational time, it yields significantly better performance. For example, compared to the best activation function on the dataset, namely Maxout, DIGRAF requires an additional $\sim 6.21$ms at inference, but results in a relative improvement in the performance of $\sim 17.95\%$.

On the ZINC dataset, using GIN as the primary model, DIGRAF exhibits approximately 4.5 times slower training times and 3.5 times slower inference times compared to ReLU. DIGRAF demonstrates an inference time that is approximately 1.35 times faster than GReLU, while also achieving superior performance.

Table 13: Hyperparameter configurations for the social network datasets [86].

| Hyperparameter | Search Range / Value |
|---|---|
| Learning rate for $\text{GNN}^{(l)}_{\text{LAYER}}$ | $[10^{-5}, 10^{-4}, 5 \times 10^{-4}, 10^{-3}, 5 \times 10^{-2}, 10^{-2}]$ |
| Learning rate for $\theta^{(l)}$ / $\text{GNN}_{\text{ACT}}$ | $[10^{-6}, 10^{-5}, 10^{-4}, 10^{-3}, 5 \times 10^{-3}, 10^{-2}, 5 \times 10^{-2}]$ |
| Weight decay for $\text{GNN}_{\text{LAYER}}$ | $[10^{-5}, 10^{-4}, 5 \times 10^{-3}, 0.0]$ |
| Weight decay for $\theta^{(l)}$ / $\text{GNN}_{\text{ACT}}$ | $[10^{-6}, 10^{-5}, 10^{-4}, 5 \times 10^{-3}, 0.0]$ |
| $C$ | $[64, 128, 256]$ |
| $C_{\text{ACT}}$ | $[16, 32, 64, 128]$ |
| $L$ | $[2, 4]$ |
| $L_{\text{ACT}}$ | $[2, 4]$ |
| $p$ | $[0.0, 0.4, 0.5, 0.6, 0.7]$ |
| $\mathcal{N}_{\mathcal{P}}$ | $[2, 4, 8, 16]$ |
| $\lambda$ | $[0.0, 10^{-3}, 10^{-2}, 1.0]$ |

Table 14: Statistics of the graph classification datasets [55, 36, 17].

| Dataset | #graphs | #nodes | #edges | #features | #classes |
|---|---|---|---|---|---|
| ZINC-12K | 12,000 | $\sim$23.2 | $\sim$49.8 | 1 | 1 |
| **TUDatasets** | | | | | |
| MUTAG | 188 | $\sim$17.9 | $\sim$39.6 | 7 | 2 |
| PROTEINS | 1,113 | $\sim$39.1 | $\sim$145.6 | 3 | 2 |
| PTC | 344 | $\sim$14.2 | $\sim$14.6 | 18 | 2 |
| NCI1 | 4,110 | $\sim$29.8 | $\sim$32.3 | 37 | 2 |
| NCI109 | 4,127 | $\sim$29.6 | $\sim$32.1 | 38 | 2 |
| **OGB** | | | | | |
| MOLESOL | 1,128 | $\sim$13.3 | $\sim$13.7 | 9 | 1 |
| MOLTOX21 | 7,831 | $\sim$18.6 | $\sim$19.3 | 9 | 2 |
| MOLBACE | 1,513 | $\sim$34.1 | $\sim$36.9 | 9 | 2 |
| MOLHIV | 41,127 | $\sim$25.5 | $\sim$27.5 | 9 | 2 |

Table 15: Hyperparameters and search ranges/values for TUDatasets [55], OGB [36], and ZINC-12K [17] datasets.

| Hyperparameter | TUDatasets | OGB | ZINC |
|---|---|---|---|
| Learning rate for $\text{GNN}^{(l)}_{\text{LAYER}}$ | $[10^{-5}, 10^{-4}, 10^{-3}, 5 \times 10^{-3}]$ | | |
| Learning rate for $\theta^{(l)}/\text{GNN}_{\text{ACT}}$ | $[5 \times 10^{-6}, 10^{-5}, 10^{-4}, 10^{-3}, 5 \times 10^{-3}]$ | | |
| Weight decay for $\text{GNN}^{(l)}_{\text{LAYER}}$ | $[10^{-5}, 10^{-4}, 5 \times 10^{-3}, 0.0]$ | | |
| Weight decay for $\theta^{(l)}/\text{GNN}_{\text{ACT}}$ | $[10^{-5}, 10^{-4}, 5 \times 10^{-3}, 0.0]$ | | |
| $C$ | $[16, 32]$ | $[64, 128]$ | $[64, 128, 256]$ |
| $C_{\text{ACT}}$ | $[16, 32, 64, 128]$ | – | – |
| $L$ | $[4, 6]$ | $[2, 4, 6]$ | $[2, 4]$ |
| $p$ | $[0.0, 0.5]$ | | |
| $\mathcal{N}_{\mathcal{P}}$ | $[2, 4, 8, 16]$ | | |
| $\lambda$ | $[0.0, 10^{-3}, 10^{-2}, 1.0]$ | | |
| Graph pooling layer | [sum, mean] | | |
| Batch size | $[32, 128]$ | $[64, 128]$ | $[64, 128]$ |

Table 16: Batch runtimes on an NVIDIA RTX A5000 GPU of DIGRAF and other activation functions, with 4 GNN layers, batch size 128, 64 embedding dimensions, and $\text{GNN}_{\text{ACT}}$ with $L_{\text{ACT}} = 2$ layers and $C_{\text{ACT}} = 64$ embedding dimension, on ZINC-12K dataset.

| Method | ZINC | | |
|---|---|---|---|
| | Training time (ms) | Inference time (ms) | (MAE $\downarrow$) |
| GIN + ReLU [84] | 4.18$\pm$0.10 | 2.47$\pm$0.08 | 0.1630$\pm$0.0040 |
| GIN + Maxout [29] | 4.71$\pm$0.13 | 2.41$\pm$0.12 | 0.1587$\pm$0.0057 |
| GIN + Swish [66] | 4.55$\pm$0.12 | 2.30$\pm$0.24 | 0.1636$\pm$0.0039 |
| GIN + Max [38] | 9.19$\pm$0.25 | 4.50$\pm$0.93 | 0.1661$\pm$0.0035 |
| GIN + Median [38] | 14.54$\pm$1.35 | 10.13$\pm$1.20 | 0.1715$\pm$0.0050 |
| GIN + GReLU [92] | 20.63$\pm$0.99 | 11.69$\pm$2.79 | 0.3003$\pm$0.0086 |
| GIN + DIGRAF (W/O ADAP.) | 13.76$\pm$0.65 | 4.97$\pm$1.72 | 0.1382$\pm$0.0080 |
| GIN + DIGRAF | 19.37$\pm$1.28 | 8.62$\pm$0.18 | 0.1302$\pm$0.0090 |

