# OpenReview forum: "DiGRAF: Diffeomorphic Graph-Adaptive Activation Function"
_NeurIPS.cc/2024/Conference — NeurIPS 2024 poster_

### Official Review · Reviewer_AwEg · 2024-07-08

**Soundness:** 3
**Presentation:** 2
**Contribution:** 3
**Rating:** 5
**Confidence:** 3

**Summary:**

Inspired by the continuous piecewise-affine based transformation, this paper argued that the activation function should not be an uniform selection for different nodes, and developed a learning activation function method, that can take the graph structure into account to activate node-specific features. The experimental results support their proposed model's effectiveness through comparing with existing activation functions in several datasets.

**Strengths:**

This paper provides a new method to learn the activation function, where instead of activating every nodes (representations) uniformly, this method helps generate a node-specific function.
The experimental results prove that this method can significantly increase the model performance.

**Weaknesses:**

Many formula seem to be repeated, such as equations in 3.2 and 4.1.
The complexity is a bit high compared with an existing activation function.

**Questions:**

1. How will you ensure the parameter \theta you learn from the training can return you a diffeomorphism activation function? It is not clearly stated in your experimental settings, such as how you define transformation T and space \Omega.
2. The method used a GNN to learn the activation function, is there an activation function in the GNN_act?
3. In the node classification experiments, there are some significant improvement in some tasks, while some of them have a minor improvement. Have you tried to figure out the reasons?

**Limitations:**

There is no potential negative societal impact of this work.

---

> ### Author Rebuttal · Authors · 2024-08-07
>
> We appreciate the reviewer's thoughtful evaluation and are happy to see that the reviewer recognized the performance increase yielded by our method. We now proceed by addressing the questions and comments, and hope you find them satisfactory to consider revising your score.
>
> **Q1**: *Many formulas seem to be repeated, such as in 3.2 and 4.1. The complexity is a bit high compared with an existing activation function.*
>
> **A1**: Thank you for the feedback. Section 3.2 serves as the blueprint of the framework, where notations, GNN layers, and a general parameterized activation function are defined. Please note that each Equations (3),(4), and (5) each describe a different tensor, and are essential to define the notations that are used throughout the paper .Section 4.1 specializes in this blueprint to diffeomorphisms, in which we explain how DiGRAF transforms the node features using the CPAB framework and how this differs from traditional activation functions. We agree that some equations in Section 4.1 can be incorporated in the text instead of having a unique Equation. One example is Equation (8) that introduces the element-wise application of DiGRAF to the input node feature tensor. We followed your guidance in our revised paper.
> Regarding the complexity of the method, please note that we discuss this aspect in the limitations paragraph in our paper. Also,  we emphasize that DiGRAF exhibits linear computational complexity with respect to the input size, and can achieve sub-linear running times through parallelization, as discussed in Section 4.3. Although DiGRAF includes an additional component compared to standard activation functions, such as ReLU and Tanh, it delivers significantly better performance, as evident from our diverse experiments in Section 5. Furthermore, compared with other graph activation functions such as GReLU we note that our DiGRAF maintains faster runtimes and superior performance. We have detailed the time complexity and measured runtimes in Table 13 at Appendix H, and compared it with other methods as well. We revised the paper to better highlight this discussion. Thank you.
>
> **Q2**: *How will you ensure the parameter $\theta$ you learn from the training can return you a diffeomorphism activation function? It is not clearly stated in your experimental settings, such as how you define transformation $T$ and space $\Omega$.*
>
> **A2**: Thank you for the question. Please allow us to start by stating that by construction, DiGRAF yields a diffeomorphic activation function within the domain $\Omega$, because of the utilization of the CPAB framework, which yields a diffeomorphic transformation based on  velocity field weights $\theta$. In the case of DiGRAF, we show that the velocity field weights can be learned to be both (i) task-aware, and (ii) graph-aware. In particular, using $\text{GNN}\_{act}$, we learn the velocity field weights $\theta$, which serves as the parameter for the diffeomorphism transformation built on the CPAB framework. Following the construction in the work of CPAB (Freifeld et al. 2017), the velocity field, computed using the parameter $\theta$ and the predefined tessellation setup, is always continuous piecewise affine (CPA). To be precise, by definition, the integral of a CPA velocity field is guaranteed to be a diffeomorphic function. Therefore, the DiGRAF learned is inherently a diffeomorphic activation function. We would like to kindly note that the mathematical background for this discussion is provided in Section 3.1, and we show how to use it to learn our graph activation function DiGRAF in Section 4. In addition, we provided an overview of CPAB transformations in Appendix B.
> Regarding the definition of $T$, as defined in Equation (1) and (6), the CPAB transformation $T$ in DiGRAF at layer $l$ is $T^{(l)}(\bar{h}^{(l)}\_{u, c}; \theta^{(l)}) \triangleq f^{\theta^{(l)}}(\bar{h}^{(l)}_{u, c}) = \phi^\theta(\bar{h}^{(l)}\_{u, c}, t=1)$, where $\bar{h}^{(l)}\_{u, c}$, defined by Equation (4), represents the intermediate value of a node between the GNN layer and the activation function layer.
> Regarding the domain $\Omega = [a, b]$, it is defined by hyperparameters $a$ and $b$ where $a < b$, as discussed in detail in Section 4. In practice, we set $a = -b$ to ensure that the activation function is symmetric and centered around 0.
> We thank you for the questions, and we revised our text to better highlight these aspects, discussed in our paper.
> Freifeld et al. 2017. Transformations based on continuous piecewise-affine velocity fields.
>
> **Q3**: *The method used a GNN to learn the activation function, is there an activation in the $\text{GNN}_{\text{act}}$?*
>
> **A3**: We use the ReLU activation function within $\text{GNN}\_{act}$, as it is  widely used in GNNs, in order to make it non-linear. The important part is that $\text{GNN}\_{act}$ predicts the velocity field that defines our learned activation function. We added these details to our paper.
>
> **Q4**: *In the node classification experiments, there are some significant improvements in some tasks, while some of them have a minor improvement. Have you tried to figure out the reasons?*
>
> **A4**: We thank the reviewer for their insightful question. Following the reviewer’s suggestion, we have conducted an investigation on the differences between the datasets. Interestingly, our findings reveal that the datasets where DiGRAF showcases the largest performance improvements  (9.5% on Blog Catalog and 18.9% on Flickr) over baselines (ReLU), have more balanced label distributions, while this is not the case for the other datasets (Cora and CiteSeer, where DiGRAF nonetheless still improves by 3.6% and 1.8%). We present the distribution of these labels as Figure 1 in the additional PDF file.
>
> We believe that further understanding the impact of the activation function and in particular of DiGRAF on imbalanced data represents an interesting future direction, which we are eager to conduct in future work.

---

### Official Review · Reviewer_SUte · 2024-07-15

**Soundness:** 3
**Presentation:** 3
**Contribution:** 3
**Rating:** 6
**Confidence:** 4

**Summary:**

This paper introduces a new activation function, DIGRAF, specifically tailored for graph data in Graph Neural Networks (GNNs). The approach is based on Continuous Piecewise-Affine Based transformation (CPAB). The authors demonstrate that DIGRAF possesses the desired properties highlighted in existing literature and provide a thorough analysis of these properties. Extensive experiments conducted on diverse datasets across various tasks show that DIGRAF outperforms three different types of baselines in downstream performance.

**Strengths:**

1. This paper devise an activation function based on CPAB, which possess many desirable properties of activation functions (e.g. zero-centered, permutation equivariant, etc.) and has solid theoretical guarantees. It's an ingenious idea and seems that this method has never been adopted before in the activation function of GNN.
2. The variables and equations are well-defined and thoroughly explained. The experiments are extensive and robust, effectively supporting the corresponding claims with comprehensive data.
3. The paper addresses the most critical questions about the effectiveness of the proposed method. Each question is answered with extensive experiments and explicit explanations, providing clear evidence for the method’s efficacy. The figures are well-designed, enhancing understanding and readability. Overall, the paper is smoothly written and easy to follow.

**Weaknesses:**

1. Apart from the adoption of the CPAB approach, the primary contribution appears to be the introduction of GNN_act. However, the discussion on why GNN_act is effective is relatively brief. Including more detailed discussions or theoretical justifications would help in understanding its advantages.
2. How do the properties of DIGRAF influence the convergence? It would be nice if it could be discussed which property the gain comes from.
3. As I mentioned above, more ablation studies should be done to reveal how each property and design contribute to the performance gain. Otherwise, DIGRAF (w/o adap.) is good and simple enough. Maybe we can save the budget of GNN_act for a larger GNN_layer.

**Questions:**

1. The size of GNN_act seems to have a minor influence (non-monotonic) on performance according to Appendix F.2. What if you increase the size of GNN_layer a little bit (add the parameter of GNN_act to GNN_layer) and compare DIGRAF (w/o adap. & larger GNN_layer) with the current DIGRAF? I believe this could provide a better justification for the effectiveness of GNN_act.
2. What are the properties that DIGRAF has but general-purpose activation functions or existing graph activation functions do not have? Can you put up a table to clarify it?
3. How do the properties of DIGRAF influence the convergence and performance?
4. The tessellation size decides the degree of freedom of DIGRAF, right?  Why does increasing the tessellation size result in marginal changes (non-monotonic)?

**Limitations:**

The authors have discussed the limitations.

---

> ### Author Rebuttal · Authors · 2024-08-07
>
> We greatly appreciate the reviewer’s constructive and positive feedback. We are delighted to see the reviewer has valued our experimental analysis. We proceed by answering the questions in the following.
>
> **Q1**: *The discussion on why $\text{GNN}_\text{act}$ is effective is relatively brief. Including more detailed discussions or theoretical justifications would help in understanding its advantages.*
>
> **A1**: We thank you for your suggestion, and mention that $\text{GNN}\_\text{act}$ boosts performance by adapting the activation function to the input graph, specifically by returning the parameter $\theta$, which governs the CPAB diffeomorphism used as an activation function. This graph adaptivity represents the critical factor of the performance boost, as evident by DiGRAF consistently outperforming DiGRAF (w/o ADAP), where $\text{GNN}\_\text{act}$ is replaced by a learnable $\theta$ which is the same for all graphs.
>
> To make this point clearer, in **Figure 2 in the additional PDF**, we plot the learned activation functions for two distinct randomly sampled graphs from the ZINC dataset, which differ in the number of nodes, features, and the connectivity. While using the non-adaptive variant DiGRAF (w/o ADAP) yields a (learned) fixed activation function for all inputs,  $\text{GNN}_\text{act}$ can account for the variables discussed above, and  learn distinct activation functions for different graphs.
>
> **Q2**: *The size of $\text{GNN}\_\text{act}$ seems to have a minor influence (non-monotonic) on performance according to Appendix F.2. What if you increase the size of $\text{GNN}\_{\text{layer}}$ a little bit (add the parameter of $\text{GNN}\_\text{act}$ to $\text{GNN}\_{\text{layer}}$) and compare DiGRAF(w/o ADAP. \& larger $\text{GNN}\_{\text{layer}}$ ) with the current DiGRAF?*
>
> **A2**: As shown in Table 5 in Appendix E, for ZINC and MOLHIV datasets the difference in the number of parameters between DiGRAF and DiGRAF (w/o ADAP) is relatively small, with DiGRAF having 10% and 30% more parameters than DiGRAF (w/o ADAP), respectively. However, we welcome the reviewer’s suggestion and we have now conducted an additional experiment, where we increase the number of parameters of DiGRAF (w/o ADAP) to match the one of DiGRAF by increasing the size of $\text{GNN}_\text{layer}$. We denote this variant by DiGRAF(w/o ADAP. \& larger $\text{GNN}\_{\text{layer}}$ ).
>
> In the Table below, we report the performance on the ZINC-12k and OGBG-MOLHIV datasets. The results show that the improvement of DiGRAF cannot be attributed to the (small) increase in the number of parameters. On the contrary, it is the way these parameters are allocated, namely for $\text{GNN}_\text{act}$, which adapts the activation function to the graph, that yields significant improvements.
>
> | Method  | ZINC (MAE) $\downarrow$  | MOLHIV (ROC AUC) $\uparrow$|
> |:--|:--|:--:|
> | DiGRAF (w/o ADAP.) with larger $\text{GNN}_\text{Layer}$ | 0.1388 $\pm$ 0.0071 (337K)   | 79.22 $\pm$ 1.40 (85K)  |
> | DiGRAF (w/o ADAP.) (Original)  | 0.1382 $\pm$ 0.0086 (308K)   | 79.19 $\pm$ 1.36 (63K)  |
> | DiGRAF   | 0.1302 $\pm$ 0.0094 (333K)   | 80.28 $\pm$ 1.44 (83K)  |
>
> Furthermore, we would like to kindly refer you to a similar experiment in our submission. In Table 5, we show that the contribution of DiGRAF (for all variants) stems from the learning of the activation function, rather than adding more parameters to a baseline model.
>
> **Q3**: *What are the properties that DiGRAF has but general-purpose activation functions or existing graph activation functions do not have? Can you put up a table to clarify it?*
>
> **A3**: We appreciate the reviewer's suggestion. We have identified the desirable properties of activation functions and examined various activation functions in the following table.
>
> | Properties \ Act | ReLU | Tanh | PReLU | Swish | Max | Median | GReLU | DiGRAF |
> |-|-|-|-|-|-|-|-|-|
> | Boundness | N | Y | N | N | N | N | N | Y (within $\Omega)$ |
> | Differentiability Everywhere | N | Y | N | Y | N | N | N | Y |
> | Linear Complexity | Y | Y | Y | Y | Y | Y* | Y | Y |
> | Permutation Equivariance | Y | Y | Y | Y | Y | Y | Y | Y |
> | Lipschitz Continuity | Y | Y | Y | Y | Y | NA in their paper | NA in their paper | Y |
> | Graph Adaptivity | N | N | N | N | Y | Y | Y | Y |
>
> *In practice, the Median-of-medians Algorithm can achieve linear time complexity on average.
>
> We discuss these properties in Section 4.3, with proofs of Lipschitz continuity and permutation equivariance provided in Appendix D. We added the Table and a discussion in our revised paper.
>
> **Q4**: *How do the properties of DIGRAF influence the convergence and performance?*
>
> **A4**: DiGRAF possesses the key properties typically associated with faster convergence: differentiability everywhere, boundedness within the input-output domain $\Omega$, zero-centrality (Szandala, 2021, Dubey et al., 2022). In our revised paper, we improved the connection between these properties and the faster convergence and performance offered by DiGRAF.
>
> **Q5**: *The tessellation size decides the degree of freedom of DiGRAF right? Why does increasing the tessellation size result in marginal changes (non-monotonic)?*
>
> **A5**: The reviewer is correct in that the tessellation size determines the degree of freedom of the velocity field. However, our ablation studies (Appendix F) demonstrate that even a small size suffices for the considered tasks. This result suggests that CPAB is highly flexible, and aligns with the conclusions in previous studies on different applications of CPAB (Martinez et al., 2022), which have shown that small sizes are sufficient in most cases.  We added this discussion to our revised paper.
>
> Martinez et al., 2022. Closed-form diffeomorphic transformations for time series alignment
>
> Dubey et al., 2022. Activation functions in deep learning: A comprehensive survey and benchmark
>
> Szandala, 2021. Review and comparison of commonly used activation functions for deep neural networks

---

### Official Review · Reviewer_LFcv · 2024-07-20

**Soundness:** 3
**Presentation:** 3
**Contribution:** 2
**Rating:** 5
**Confidence:** 4

**Summary:**

This paper introduces DIGRAF, Diffeomorphic Graph-Adaptive Activation Function, which is a novel activation function adaptive to graph data. DIGRAF leverages Continuous Piecewise-Affine Based transformations, possesses several necessary properties of a good activation function, such as differentiability, zero-centering and permutation equivariance. The author allows DIGRAF adapt to graph data by using an additional GNN to learn the final activation function formula. Adequate experiments which take use of DIGRAF in GNNs proved the high efficiency of this activation function.

**Strengths:**

1.	The activation function DIGRAF is a good choice for GNN model design, more effective than the common activation function such as ReLU.
2.	The soundness of this paper is wonderful. The theoretical proof of DIGRAF’s properties is detailed and experiments in this paper is convictive, which contain node classification, graph classification and graph regression in task level.

**Weaknesses:**

There should be an ablation study (can be set in the supplementary part) about the hyper-parameters and training strategy in the process of leaning DIGRAF.

**Questions:**

1. In the experiments, why was the sigmoid function not used as an activation function?
2. Are there any differences in the DIGRAF learning process for GCN and GIN?

**Limitations:**

1. As the author mentioned, the current DIGRAF may not be optimal since it was learned only for simple tasks, but it can be improved in future work.
2. The application of DIGRAF has not been discussed or implemented in this research. However, using DIGRAF as a novel activation function in some current models might improve their efficiency.

---

> ### Author Rebuttal · Authors · 2024-08-07
>
> We are glad to see the reviewer has particularly appreciated the soundness of our paper, while finding the theoretical analysis detailed and the experiments convincing. We proceed by answering the questions raised by the reviewer in the following. We hope that you find our responses satisfactory to consider revising your score. We are welcoming any question, comment, or suggestion.
>
>
> **Q1**: *There should be an ablation study (can be set in the supplementary part) about the hyper-parameters and training strategy in the process of learning DiGRAF.*
>
> **A1**: We would like to kindly note that Appendix F contains an ablation study on the effect of the hyper-parameters added by DiGRAF. We found that a small tessellation size is sufficient for good performance, and increasing its size results in only marginal changes (Figure 7). Moreover, increasing the depth of $\text{GNN}_\text{act}$ improves the performance of DiGRAF marginally (Table 6). For regularization, the results (Table 7) reveal that the optimal value of the coefficient depends on the dataset of interest, with small positive values generally yielding good results across all datasets. Also, we discuss the training loss, protocols and hyperparameter tuning in Appendices B and G, respectively. In our revised paper, we improved the link between the relevant parts in the main paper and the appendices. Thank you.
>
> **Q2**: *In the experiments, why was the Sigmoid function not used as an activation function?*
>
> **A2**: We appreciate the reviewer's observation. We have conducted additional experiments using the Sigmoid function, and added them to our revised paper. As can be seen from the table below, DiGRAF consistently outperforms Sigmoid with a large margin across various datasets:
>
> |Activation\Dataset| Blog Catalog $\uparrow$  | Flickr $\uparrow$ | CiteSeer $\uparrow$ | Cora $\uparrow$ | PubMed $\uparrow$ |
> |-|-|-|-|-|-|
> | Sigmoid |39.7 $\pm$ 4.5| 18.3 $\pm$ 1.2| 27.9 $\pm$ 2.1 | 32.1 $\pm$ 2.3 | 52.8 $\pm$ 6.6 |
> | DiGRAF (w/o ADAP.) | 80.8 $\pm$ 0.6 | 68.6 $\pm$ 1.8 | 69.2 $\pm$ 2.1 | 81.5 $\pm$ 1.1 | 78.3 $\pm$ 1.6 |
> | DiGRAF | 81.6 $\pm$ 0.8 | 69.6 $\pm$ 0.6 | 69.5 $\pm$ 1.4 | 82.8 $\pm$ 1.1 | 79.3 $\pm$ 1.4|
>
>
> |Activation\Dataset|  MUTAG $\uparrow$  | PTC $\uparrow$ | PROTEINS $\uparrow$ | NCI1 $\uparrow$ |  NCI109 $\uparrow$ |
> |-|-|-|-|-|-|
> | Sigmoid | 90.9 $\pm$ 5.5 | 65.3 $\pm$ 4.8 | 75.0 $\pm$ 5.0 | 82.6 $\pm$ 1.4 | 81.2 $\pm$ 1.6 |
> | DiGRAF (w/o ADAP.) | 92.0 $\pm$ 5.6 | 68.9 $\pm$ 7.5 | 77.2 $\pm$ 3.6 | 83.0 $\pm$ 1.3 | 82.9 $\pm$ 2.2 |
> | DiGRAF | 92.1 $\pm$ 7.9 | 68.6 $\pm$ 7.4 | 77.9 $\pm$ 3.4 | 83.4 $\pm$ 1.2 | 83.3 $\pm$ 1.9 |
>
>
> |Activation\Dataset| MOLESOL (RMSE) $\downarrow$| MOLTOX21 (ROC AUC) $\uparrow$ | MOLBACE (ROC AUC) $\uparrow$ | MOLHIV (ROC AUC) $\uparrow$ |
> |-|-|-|-|-|
> | Sigmoid | $0.8836 \pm 0.043$ | $69.15 \pm 0.52$ | $68.70 \pm 3.68$ | $73.87 \pm 0.80$ |
> | DiGRAF (w/o ADAP.) | $0.9011 \pm 0.047$ | $76.37 \pm 0.49$ | $78.90 \pm 1.41$ | $79.19 \pm 1.36$ |
> | DiGRAF | $0.8196 \pm 0.051$ | $77.03 \pm 0.59$ | $80.37 \pm 1.37$ | $80.28 \pm 1.44$ |
>
>
> |Activation\Dataset|  ZINC (MAE) $\downarrow$  |
> |-|-|
> | Sigmoid | $0.3839 \pm 0.0058$ |
> | DiGRAF (w/o ADAP) | $0.1382 \pm 0.0080$ |
> | DiGRAF | $0.1302 \pm 0.0090$ |
>
>
>
> **Q3**: *Are there any differences in the DiGRAF learning process for GCN and GIN?*
>
> **A3**: There are no differences in the learning process for DiGRAF in GCN and GIN. Specifically, although the GNN layer in $\text{GNN}\_{act}$ aligns with the primary GNN model, $\text{GNN}\_{act}$ still takes in the input graph data and returns the parameter $\theta$ that is used in the activation DiGRAF. In terms of training, we did not change the training procedure for different backbones. We added this important discussion to the revised paper. Thank you.
>
> **Q4**: As the author mentioned, the current DIGRAF may not be optimal since it was learned only for simple tasks, but it can be improved in future work.
>
> **A4**: We thank the reviewer for their comments. While we evaluated DiGRAF on a wide variety of available real-world datasets and tasks, which are widely utilized by the GNN community, it would be interesting to see DiGRAF performance on additional real-world tasks, and we are eager to explore this in future work. Our motivation in demonstrating DiGRAF on such a variety of datasets and benchmarks is to show its general effectiveness, and its ability to cater different communities that utilize GNNs to solve hard problems like drug discovery and weather prediction. Overall, we found that DiGRAF consistently offers better performance than other activation functions, thereby motivating its application in additional tasks. We thank you for the insightful comment, and we added this discussion to our revised paper.
>
> **Q5**: *The application of DiGRAF has not been discussed or implemented in this research. However, using DiGRAF as a novel activation function in some current models might improve their efficiency.*
>
> **A5**: We believe the reviewer is referring to applying an adaptive activation function, such as DiGRAF, to other domains. While this represents an intriguing avenue of research, it falls outside the scope of the current work, which specifically focuses on graph inputs, as discussed throughout our paper. DiGRAF leverages graph information, making it particularly well-suited for graph-related tasks. We believe however specialization to other domains is possible and we are happy to try in future work. It is also important for us to note that to comprehensively demonstrate the effectiveness of DiGRAF, our experimental section was carefully designed to include different tasks, from node to graph level, inductive and transductive, spanning across various domains, from citation networks to molecular datasets. We further highlighted these aspects in our revised paper - thank you.

---

### Official Review · Reviewer_mZzM · 2024-07-23

**Soundness:** 3
**Presentation:** 3
**Contribution:** 2
**Rating:** 5
**Confidence:** 3

**Summary:**

The paper “DiGRAF: Diffeomorphic Graph Activation Functions” introduces DiGRAF, a novel graph neural network (GNN) activation function based on diffeomorphisms. DiGRAF adapts to graph structures and tasks by learning transformation parameters, enhancing performance across various GNN scenarios. The method demonstrates its effectiveness through extensive experiments on node classification, graph classification, and regression tasks, showing significant improvements over standard and graph-specific activation functions.

**Strengths:**

1.	Power of DiGRAF: The paper effectively demonstrates the superiority of DiGRAF in various GNN scenarios, showcasing its adaptability and performance improvements.
2.	Extensive Experiment Analysis: The experiments are comprehensive, covering multiple datasets and comparing them with various baseline activation functions.
3.	Learnable Activation Function: DiGRAF uses a flexible, learnable activation function that is differentiable and bounded, providing desirable characteristics for machine learning applications.

**Weaknesses:**

1.      Limited Contribution: The paper appears to be more of an application of CPAB. The primary contribution of the analysis of the connection between CPAB and GNNs is not thoroughly explored.
2.	Performance and Gradient Optimization: The paper lacks a detailed analysis of performance and gradient optimization aspects.
3.	Clarification on GNN_act: It needs to be clearer how GNN_act helps boost performance. Is there a way to analyze the new source of information it brings?
4.	Proofs for GNN: Most analyses focus on the boundedness of the activation function without specifics tailored to GNNs, making it unclear why this approach is particularly suited for GNNs.
5.	Edge Classification Performance: The report on edge classification performance is missing.
6.  	Hyper-Parameter Effects: It is not clear what the effect of each hyper-parameter is on the performance of the activation function.
7.   	Source Code: The source code is not available.

**Questions:**

How are the parameters {b_j} (orthonormal basis of the space of velocity fields V) calculated?
2.	What would be the activation function of GNN_act if the method’s purpose was to design a new activation function?
3.	Are there results for other GNN structures beyond GCN? For instance, GReLU reports results for different GNN structures.
4.	Since DiGRAF is also applicable to non-graph datasets, would it be beneficial to see its results on other machine learning methods?
5.	What is the closed-form solution of the integral in Equation (2) for 1-dimensional vectors?
6.	Is there any information on the learned \theta parameters in terms of sparsity or graph structure?

**Limitations:**

Inductive Learning: Since the method is task-dependent, it loses its inductive capabilities and cannot be used for cases it did not see.
2.	Hyperparameter Choice: There is a dependency on hyperparameter tuning, which may affect generalizability.
3.	Time Complexity: DiGRAF has higher time complexity compared to standard activation functions.
4.	Decentralized Tasks: The approach is not ideal for decentralized tasks where there is no central data.

---

> ### Author Rebuttal · Authors · 2024-08-07
>
> We sincerely thank the reviewer for the insightful comments, and appreciation of our experiment analysis. We now address them. New results and discussions were added to our paper.
>
> **1. On Contribution**: We distinguish between CPAB's original purpose (signal alignment via diffeomorphism) and our novel use for learning an activation function (Section 4.1 and Figure 4). Additionally, we made several key contributions: (i) Sections 4.2 and 5 demonstrate the importance of correctly parameterizing the velocity field for graph adaptivity, (ii) Section 4.3 discusses DiGRAF's properties, justifying its design and effectiveness as a graph activation function.
>
> **2.On Optimization**: DiGRAF's behavior is analyzed by its Lipschitz constant (Proposition D.2), related to its optimization[5,6]. Figure 5 and Appendix E.1 show DiGRAF's faster convergence and lower loss compared to other activation functions.
>
> **3.On $\text{GNN}_\text{act}$**:  $\text{GNN}_\text{act}$ adapts the activation function to the graph, by returning the parameter $\theta$ for the learned diffeomorphism. This graph adaptivity is crucial, as shown by DiGRAF consistently outperforming its non-adaptive counterpart (DiGRAF w/o ADAP).
>
> Figure 2 in the additional PDF shows learned activation functions for two graphs from ZINC, differing in number of nodes, features, and connectivity. $\text{GNN}_\text{act}$ enables DiGRAF to capture these differences and learn distinct activation functions for them.
>
> **4.On DiGRAF Analyses**: DiGRAF is well-suited for GNNs because $\theta$, determining the activation function (Equation 7), is learned by $\text{GNN}_{\text{act}}$ (Equation 9), making it adaptive to different input graphs. Our experiments and response 3 highlight that adaptivity enhances performance.
>
> **5.On Edge Classification**:  Following your suggestion, we performed experiments on link prediction using GCN+DiGRAF on the OGBL-Collab dataset (**Table 2, Additional PDF**), showing that DiGRAF improves ReLU and surpasses DiGRAF (w/o ADAP) on this additional task, supporting our claims of consistent improvements.
>
> **6.On Hyper-parameters**:  We provide experiments on key hyper-parameters (Appendix F). A small tessellation size suffices for good performance (Figure 7). Increasing $\text{GNN}_\text{act}$ depth marginally improves DiGRAF's performance (Table 6). Regularization coefficient $\lambda$ varies by dataset, with small positive values yielding good results (Table 7).
>
> **7. On Code**: As promised in our submission, we will publicly release the code upon acceptance.
>
> **8. On $b_j$**:The orthonormal basis $\mathbf{B}$ for the velocity field is obtained via SVD of $\mathbf{L}$, which constrains velocity function coefficients by ensuring consistent values at shared endpoints [4].
>
> **9. On Activation in $\text{GNN}_\text{act}$**: We use ReLU within $\text{GNN}_\text{act}$, as it is  widely used in GNNs, to make it non-linear.
>
> **10. On GNN Structures**: Our experiments include GCN and GIN. With your suggestion, we additionally tested GAT [1] and SAGE [2]. Table 1 in the additional PDF shows that DiGRAF continues to consistently offer superior results.
>
> **11. On Non-graph data**: Graphs are more diverse than other data due to their unstructured nature, making adaptativity more crucial. Still, DiGRAF can potentially be specialized to other domains, like vision, by using a CNN instead of $\text{GNN}_\text{act}$ or our non-adaptive variant. We are eager to explore this in future work.
>
> **12. On Closed-form solution**: Eq. (2) has an equivalent ODE [4]. By varying $x$ and fixing $t$ the solution to this ODE can be written as a composition of a finite number of solutions:
> $\\phi^{\theta} (x, t) = (\psi^{t_m}\_{\theta, c_m} \circ \psi^{t_{m-1}}\_{\theta, c_{m-1}} \circ \cdots \circ \psi^{t_2}\_{\theta, c_2} \circ \psi^{t_1}_{\theta, c_1})(x)$
>
> Here $m$ is the number of cells visited. Given $x, \theta$, time $t$, and the smallest cell index containing $x$, $c$, we can compute each $\psi^{t_i}\_{\theta, c_i}(x), i \in \{1, …, m\}$ from $\psi^{t_1}\_{\theta,c_1}(x)$ to $\psi^{t_m}_{\theta,c_m}(x)$.
> This iterations continue until convergence, with an upper bound for $m$ being $max(c_1, N_P-c_1+1)$, where $c_1$ is to the first visited cell index, and $N_P$ is the number of closed intervals in $\Omega$. Unrolling these steps, we obtain the closed form solution for Eq. (2).
>
> **13. On $\theta$**: $\theta$ governs the learned velocity field for the activation function. While CPAB transformations don’t require $\theta$ to be sparse, the sparsest vector (zeros) results in an identity activation function. Predicted by $\text{GNN}_\text{act}$, $\theta$ reflects the graph structure and can produce different functions for different graphs, as discussed in response 3.
>
> **14. On Inductive Learning**: DiGRAF is inductive in the sense of graph learning, it is capable of handling new graphs in tests. By design, it is task and data-driven, which may limit generalization to entirely new tasks. This is common to almost all GNN models, and not specific to DiGRAF. Recent Graph Foundation Models study generalization to new tasks, and it would be interesting to adapt DiGRAF to such models.
>
> **15. On Hyperparameter Tuning**: Tuning is common in GNNs [3]. Our experiments show DiGRAF's effectiveness across diverse tasks and datasets. The ablation study (Appendix F) demonstrates that DiGRAF performs consistently in different hyperparameters.
>
> **16. On Complexity**: While DiGRAF requires more computations than ReLU, its asymptotic complexity is still linear (Section 4.3). Table 13 shows DiGRAF is ~3.5x slower than ReLU but offers significantly better performance. Additionally, DiGRAF is 1.35x faster than other graph activation functions and achieves better performance.
>
> **17. On Decentralized Tasks**: This limitation is common to most GNNs, similar to the discussion in 14. While it is an interesting research direction, addressing it is beyond the scope of our work.

---

> > ### Comment · Reviewer_mZzM · 2024-08-13
> >
> > Thanks for your response. I read it carefully.

---

### Author Rebuttal · Authors · 2024-08-07

# General Response

We would like to express our gratitude to all reviewers for their valuable feedback.
Overall, the reviewers appreciated the breadth and depth of our experimental analysis, defined *``extensive``* (**mZzm, SUte**) and *``comprehensive``* (**mZzm**). They found our experimental design to be *``robust``* (**SUte**), and *``adequate and convincing``* (**LFcv**). They recognized the *``significant improvements``*  (**mZzm**) over standard and graph-specific activation functions, acknowledging our method *``can significantly increase``* (**AwEg** ) the performance of GNNs.

We are also pleased to see that reviewer **LFcv** highlighted the soundness of our contributions, describing our theoretical proof as *``detailed``*.  Furthermore, reviewer **SUte** commented on the clarity of our presentation, noting that our notations and equations are *``well-defined and thoroughly explained``* with *``well-designed figures``*, and emphasized that our method addresses the *``most critical questions``* effectively (**SUte**).

Your thoughtful comments and suggestions allowed us to improve our paper, and we provided individual responses to each reviewer. We hope that you will find them satisfactory, and that you will consider revising your score. We are happy to discuss existing or additional questions and suggestions you may have.

**New Experiments.** Several additional experiments were conducted following the reviewers’ comments, as follows:

1. A visualization of the learned activation functions for two randomly chosen graphs from the ZINC dataset, showing how $\text{GNN}_\text{act}$ adapts to different graph structures and results in different activation functions (**mZzm, SUte**);

2. An experiment on an additional task, namely link prediction, using GCN and comparing ReLU, DiGRAF (w/o ADAP) and DiGRAF, further demonstrating DiGRAF’s  consistent improvements (**mZzm**);

3. Additional experiments using different GNN backbones beyond GCN and GINE, such as GAT and SAGE, showing DiGRAF consistently yields performance improvements, regardless of the base GNN architecture (**mZzm**), further highlighting the effectiveness of DiGRAF across multiple benchmarks and its applicability to multiple GNN backbones;

4. Additional baselines using Sigmoid as the activation function on all tasks and datasets (**LFcv**), which we show to be consistently outperformed by our DiGRAF;

5. A comparison of the performance between DiGRAF (w/o ADAP) and DiGRAF using the same number of parameters (**SUte**). Our results in this experiment further support our findings in the paper, and in particular in Table 5, showing that the performance improvement of DiGRAF cannot be attributed to the (small) increase in the number of parameters, but rather in the way in which they are allocated to obtain graph-adaptivity.

All new experiments were discussed in their respective responses, as well as in the added rebuttal PDF. We also added all discussions and results to our revised paper.

***
References:

[1] Veličković et al., 2018. Graph Attention Networks

[2] Hamilton et al., 2017. Inductive Representation Learning on Large Graphs

[3] Tönshoff et al., 2023. Where did the gap go? reassessing the long-range graph benchmark

[4] Martinez et al., 2022. Closed-form diffeomorphic transformations for time series alignment

[5] Scaman and Virmaux, 2018. Lipschitz regularity of deep neural networks: analysis and efficient estimation

[6] Xu and Zhang, 2023. Uniform Convergence of Deep Neural Networks with Lipschitz Continuous Activation Functions and Variable Widths

---

### Author Response · Authors · 2024-08-12

Dear Reviewers,

We thank you for the effort in reviewing our paper, and the overall positive feedback. \
We are also thankful for the insightful questions, comments, and suggestions, that helped us improve the paper.

As the rebuttal period ends soon, we wanted to reach out and check if you feel that our rebuttal answers your questions. \
We have worked to thoroughly address your comments, and our rebuttal includes clarifications and discussion to important comments raised by the reviewers, as well as additional experimental results as described in our General Response.\
All of the additional results and discussions were also added to our revised paper.

Please, kindly let us know of any further questions you might have. We look forward to the discussion with you.

Thank you, and best regards,\
Authors.

---

### Decision · Program_Chairs · 2024-09-25

**Decision:**

Accept (poster)

**Comment:**

The contribution of this paper consists of a novel activation function in learning by GNNs.  Specifically they introduce an approach, referred to as DIGRAF exploiting Continuous Piecewise-Affine Based (CPAB) transformations augmented with an additional GNN to learn a graph-adaptive diffeomorphic activation function in an end-to-end set up. In addition to  graph-adaptivity and flexibility, DIGRAFwas show to enjoy  desirable  properties for activation functions, including  differentiability, boundness. The authors have  run experiments to demonstrate a consistent and superior performance of DIGRAF relative to traditional and graph-specific activation functions.
All reviews but one who recommended weak accept, have recommended borderline accept.
Most of the reviewers had recommended additional experiments (ablation, more baseline and other) and the authors have for the most part  complied and have provided additional experimental results.
The most negative review stated that the real  contribution, namely  the analysis  of the connection between CPAB and GNNs needed further theoretical exploration, which, while valid, raises the question of how much can be included in a given paper.
The second comment which is also a valid one is the non-inclusion of the code for the approach, which the authors respond to, by promising to publish the code after the paper is accepted, which is clearly enforceable.
Overall the reviews were predominantly positive.